# Trace elements in fish species from the Punjnad headworks: Bioaccumulation and human health risk assessment

Saima Naz[1]*, Qudrat Ullah[2]*, Dalia Fouad[3], Abdul Qadeer[4], Maria Lateef[1], Muhammad Waqar Hassan[5], Ahmad Manan Mustafa Chatha[5]

1 Department of Zoology, Government Sadiq College Women University, Bahawalpur, Pakistan,
2 Department of Theriogenology, Faculty of Veterinary and Animal Sciences, Cholistan University of Veterinary and Animal Sciences, Bahawalpur, Punjab, Pakistan, 3 Department of Zoology, College of Science, King Saud University, Riyadh, Saudi Arabia, 4 Department of Cell Biology, School of Life Sciences, Central South University, Changsha, China, 5 Department of Entomology, Faculty of Agriculture and Environment, The Islamia University of Bahawalpur, Bahawalpur, Pakistan

* qudratullah1@cuvas.edu.pk (QU); saima.naz@gscwu.edu.pk (SN)

## Abstract

Aquatic toxicology, as a result of industrial and agrieqcultural effluences, has become a global concern impacting not only the well-being of aquatic organisms but human health as well. The current study evaluated the impact of four toxic trace elements (TTEs) Cadmium (Cd), copper (Cu), lead (Pb), and nickel (Ni) in three organs (liver, gills, and muscles) of five fish species viz, *Rita rita*, *Sperata sarwari*, *Wallago attu*, *Mastacembelus armatus*, and *Cirrhinus mrigala* collected from right and left banks of Punjnad headworks during winter, spring, and summer. We investigated the accumulation (mg/kg) of these TTEs in fish in addition to the human health risk assessment. The obtained results showed that *W. attu* accumulated significantly more TTEs (p < 0.00) as compared to other fish. Among seasons, summer had significantly more (p < 0.00) accumulation of TTEs than other seasons. Lead (Pb) accumulation was highest across TTEs in fish liver as compared to gills and muscles. The right bank showed higher accumulation (p < 0.00) of all TTEs in all fish species compared to the left bank. The human health risk assessment showed that Cd and Pb had higher exposure levels than Cu and Ni. Furthermore, the THQ was in the order of Cd > Pb > Ni > Cu. All fish species had THQ 1 for Cd and Pb and TTHQ > 1 for all fish. MPI index showed moderate to high levels of TTE contamination in all fish species. The study concluded that the right bank has higher metal accumulation than the left bank. However, fish consumption from both study sites was unsafe for human consumption. Further studies are required to evaluate the contamination of other trace elements in the aquatic ecosystem of the current site. This study will be useful for policymakers and the water department to take necessary counteractions to reduce the impact of TTEs at the study site.

**Data Availability Statement:** All relevant data are within the manuscript and its Supporting Information files.

**Funding:** The author(s) received no specific funding for this work.

**Competing interests:** The authors have declared that no competing interests exist.

## 1. Introduction

The physical, chemical, and biological characteristics of water, specifically its suitability for a specific purpose, such as drinking, swimming, agriculture, or supporting aquatic life, is referred to as Water quality [1, 2]. It is measured by the levels of organic and inorganic chemicals in the water, along with certain physical properties [3]. Natural water can become contaminated by untreated waste from industries, agriculture, and technology, which often contain metallic compounds in traces [4]. These trace elements are particularly harmful because they do not break down naturally [5, 6], can build up in food chains [7], and can disrupt aquatic ecosystems and the organisms living in them, thus becoming toxic for living organisms [8]. These toxic trace elements (TTEs) enter into the water through waste from industries like tanneries, textiles, metal finishing, mining, dyeing, ceramics, and pharmaceuticals [9]. When fish absorb TTEs, they can pass them on to humans through the food chain, leading to serious health risks, including potentially fatal consequences [6, 10, 11].

The accumulation of TTEs in fish mainly depends upon the concentration of these metals in the aquatic environment and the duration of exposure. However, several other factors, such as pH, water salinity, hardness, temperature, fish size and age, ecological needs, life cycle, capture season, and feeding habits, also significantly influence metal accumulation [8]. When metal levels in fish tissues become excessively high, they can become toxic [12, 13]. Toxic trace elements are stable and persistent contaminants in aquatic environments. Some trace elements, like zinc (Zn), copper (Cu), iron (Fe), and manganese (Mn), are essential for metabolic processes in organisms but exist in a narrow range between being beneficial and toxic [14]. Other trace elements, such as cadmium (Cd), mercury (Hg), chromium (Cr), and lead (Pb), can be extremely toxic even at low concentrations, especially under certain conditions, making regular monitoring of sensitive aquatic environments necessary [15]. The process of bio-magnification can cause pollutants to reach toxic levels in species higher up the food chain, particularly in freshwater systems [16, 17].

Fish, which occupy a critical position in the aquatic food chain, are particularly sensitive to pollution caused by TTEs. For instance, Prolonged exposure to Cd can lead to its accumulation in various tissues of fish, affecting the structure and function of vital organs like the gills, liver, and gonads [18]. The use of cadmium-containing fertilizers, agricultural chemicals, pesticides, and sewage sludge on farmland can contribute to water contamination [19]. Likewise, When fish are exposed to Copper (Cu), notable changes in the liver include hepatocyte vacuolization, necrosis, shrinkage, nuclear pyknosis, and an increase in sinusoidal spaces were noticed [20]. Copper is highly toxic in aquatic environments, affecting fish, invertebrates, and amphibians even at low environmental concentrations [21]. In fish, it tends to accumulate in various organs of fish like liver, kidney, gills, bones, fins, and muscles [22, 23]. Lead (Pb) is a toxic trace element that causes a wide range of adverse health effects, which vary depending on the dose. Both fish and humans are primarily exposed to lead through ingestion and inhalation [24]. Lead tends to accumulate in muscles, bones, blood, and fat, making it a potent environmental pollutant with significant implications for human health [25, 26]. Nickel (Ni) is another trace metal widely distributed in the environment, released from both natural sources and human activities, including stationary and mobile sources. It can be found in the air, water, soil, and biological materials [27, 28]. In fish, exposure to high concentrations of Ni has been linked to higher mortality rates for showing similar sensitivity to chronic toxicity.

The Mrigal fish, like *Cirrhinus mrigala* are widely used as food and hold significant economic importance in its native regions [29]. *Cirrhinus mrigala* is primarily herbivorous, with a diet consisting of plant matter, including algae, aquatic plants, and detritus. However, it may also consume small invertebrates. *Wallago attu*, a large freshwater catfish, is found across

rivers, reservoirs, and connected watersheds in the Indian subcontinent, including Pakistan [30]. Its rapid growth elongated silvery body, and high nutritional quality have made it a focus of aquaculture development. It is primarily carnivorous and feeds on a variety of prey, including small fish, insects, and crustaceans. The declining wild populations of *Wallago attu* have led to its classification as an endangered species [31]. Increased consumer demand has driven the development of intensive aquaculture for this species in Asian countries [32]. The spiny eel, *Mastacembelus armatus*, is one of the most common and economically significant inland teleost species in Asia, known for its high market and nutritional value [33]. It is a popular table fish due to its delicious taste and high nutritional content, and it is also popular as an aquarium fish. Like *W. attu*, this fish is also carnivorous and feeds on a variety of prey, including small fish, insects, and crustaceans. Recently, it has gained attention as an indigenous ornamental fish exported from India to other countries [34]. The Indus catfish, *Sperata sarwari*, naturally inhabits a variety of freshwater bodies across South Asia, from Afghanistan to Thailand [35]. It primarily lives in riverine habitats and feeds on various prey like small fish, insects, and crustaceans. It can also survive and breed in ponds, lakes, tanks, channels, and reservoirs [34]. *Rita rita*, a freshwater catfish of the family Bagridae, inhabits tropical rivers and estuaries and is an important food fish with high nutritional value. It is also a carnivorous fish and is commonly used as a bioindicator for monitoring riverine pollution [36]. Known for its hardiness and tolerance to wide fluctuations in water quality due to human activities, *R. rita* has recently become a key species in aquatic pollution monitoring and biomarker response studies [36].

The study of the accumulation of TTEs at Punjnad headworks and its banks is mostly neglected as only a few studies have reported bioaccumulation of TTEs at the Punjnad headworks [2]. However, to date, no scientific literature is available on the effects of TTEs on fish or other aquatic organisms in the left bank (LB) and right bank (RB) of Punjand headworks. Understanding how various TTEs bioaccumulate in fish is vital, as fish are a major food source in the region, and contamination could directly impact public health. This study aims to provide insights that can contribute to environmental management practices and inform local policies on water and food safety, ensuring the long-term well-being of the ecosystem and the community that depends on it. The present study evaluated the effects of four TTEs (Cd, Cu, Ob, and Ni) being more toxic and in higher concentrations [2] on the left bank (LB) and right bank (RB) of Punjnad headworks. The study explored the accumulation of selected TTEs in three organs (liver, gills, and muscles) of five fish species (*R. rita*, *S. sarwari*, *W. attu*, *M. armatus*, and *C. mrigala*, during winter, spring, and summer collected from Left and Right banks of Punjnad headworks. Furthermore, the health risk assessment of these TTEs to human health was also determined to explore their toxic effects on mature and juvenile consumers. This study will be useful to provide a basis and informative resource for various stakeholders like the water department, policymakers, fishermen, and fish consumers. The study provides valuable data that can help policymakers develop stricter regulations on water pollution and fish consumption advisories. Fishermen can use this information to make informed decisions about safe fishing zones, while consumers can be more aware of the potential health risks associated with contaminated fish.

## 2. Materials and methods

### 2.1 Study area and fish sampling

**2.1.1 Permissions obtained for the study.** The sampling site is a protected area under the supervision of the Punjab Fisheries Department, Punjab, Pakistan. Permission was obtained from the Department of Fisheries, Punjab, Pakistan prior to the study to collect fish and water

samples from the study site. During the entire duration of sampling, a departmental representative was accompanied.

**2.1.2 Protocol used to alleviate suffering during the sacrifice.** It is important to use an approved protocol to sacrifice the sampled fish to alleviate suffering during sacrifice. The current study used a Rapid cooling method to sacrifice the fish sample. For this purpose, the fish were immersed in an ice-water mixture (1–4˚C), inducing rapid cold shock, causing unconsciousness and then death [37].

**2.1.3 Study site.** The Punjnad headworks, located near Uch Sharif in Punjab, Pakistan, is a crucial agricultural region where the five rivers (Beas, Sutlej, Ravi, Chenab, and Jhelum) of Punjab merge into one river, namely Chenab. This area is vital for meeting the irrigation demands of the various districts of Bahawalpur and Rahim Yar Khan, as well as the northern parts of Sindh. It is characterized by extensive industrial and agricultural activities, which significantly contribute to the accumulation of various TTEs in the river, Chenab, and Punjnad headworks. The current study was conducted at left bank (LB) (Latitude (29˚23'01.0"N); Longitude (71˚05'34.8E)), and right bank (RB) (Latitude (29˚23'46.9N); Longitude (71˚01'48.7E)), of Punjnad headworks, Uch Sharif, Punjab, Pakistan (**Fig 1**).

The selected study sites were 5km apart from one another. The river banks were selected as no previous work has been reported on the accumulation of TTEs at these sites. The research specifically focuses on riverbanks because considering seasonal variations in trace metals is crucial for a comprehensive investigation. Contaminated irrigation water can significantly impact crops grown along the riverbanks, posing serious risks to agricultural productivity and food safety [38].

**2.1.4 Study site.** The study was designed to check the deleterious effects of nickel (Ni), chromium (Cr), copper (Cu), and lead (Pb) in organs (liver, gills, and Muscles) of five freshwater fish species. Due to the limited resources, the four most toxic trace elements (Cd, Cu, Pb, and Ni) were included in the study. A sampling of five fish species viz, *R. rita*, *S. sarwari*, *W. attu*, *M. armatus*, and *C. mrigala* was carried out once a month from November 2021 to July 2022. Fish species were selected based on their abundance and economic value in the study area. The selected species were most abundant in the riverbanks and were mostly preferred by the local consumers. Three fish samples for each fish species were collected during each sampling. A total of 27 fish (between the age of 6 months to 18 months) samples were collected during nine sampling months. The sampling months were divided into three sampling seasons (winter, spring, and summer). Samples from each species were randomly collected once a month during the day from study sites using a gauze net measuring 100 m × 6 m, with a mesh size of 60 mm, with the help of local fishermen. For sampling, non-metallic sampling equipment made of nylon was used to avoid metallic contamination. Disposable powder-free gloves were used for handling fish samples during storage to prevent any contamination. After collection, samples were cleaned with clean, deionized water to avoid transfer of contaminants between samples. After collection, the samples were placed inside labeled polythene bags and were immediately stored in a storage box (Coleman 48 Quart icebox) with crushed dry ice to keep the samples fresh and then transported to the Zoology Laboratory at Government Sadiq College Women University, Bahawalpur. The average weight (g), average fork length (cm), and average total length (cm) of the fish were recorded for all sample fish specimens (**S1 Table**). In the laboratory, the samples were stored until further analysis at −20˚C in a freezer (Haier −25˚C biomedical) to preserve tissue integrity and avoid decomposition, which can affect heavy metal concentrations. The initial identification and common names of the fish species were verified with the assistance of local fish catchers and sellers. An expert taxonomist used systematic keys to accurately identify all species and correct any misidentifications [39].

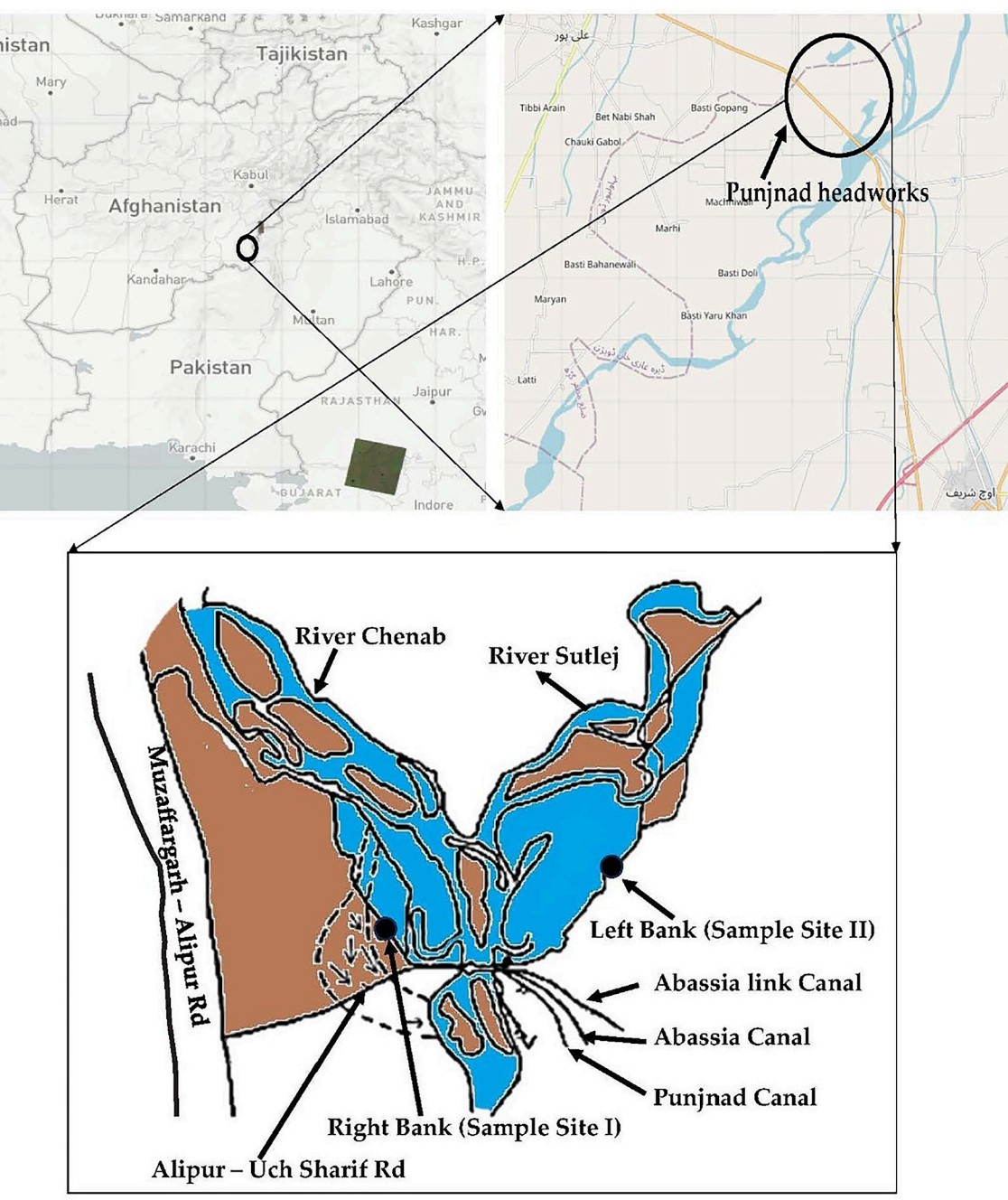

Imagery from OpenAerialMap, licensed under CC BY 4.0. https://openaerialmap.org
OpenStreetMap® is open data, licensed under the Open Data Commons Open
Database License (ODbL) by the OpenStreetMap Foundation (OSMF).

**Fig 1. Study area at the right bank (RB) and left bank (LB) of Punjand headworks, Uch Sharif, Punjab, Pakistan.** The Sampling sites are indicated by (●).

**2.1.5 Ethics approval and consent to participate.** This study was approved by the Department Committee on Animal Ethics and Welfare, Government Sadiq College Women's University, Bahawalpur, Pakistan.

## 2.2 Water samples and physicochemical properties of water at the study sites

Water samples (from 1.5 m depth with the help of a water sampler) were taken from the fish sampling sites (LB and RB) for monitoring the level of metals from Punjnad headworks, Punjab, Pakistan. The sites were covered uniformly, and sampling will be done three times a year from November 2021 to July 2022. Water samples were collected in leak proof plastic bottles that were precleaned with distilled water and preserved, as per standard practices. Some of the physicochemical parameters include the electrical conductivity of the water samples (Model S-611L, by Peak Instruments US), pH (Model STARTER300, OHAUS corporation, USA), and water temperature (by glass thermometer). The other water criteria, including total dissolved solids- TDS (Model- BANTE-510, BANTE Instruments, USA) and dissolved oxygen- DO (Model- DO200A, YSI, USA), were measured according to the traditional manual methods.

## 2.3 Digestion and preparation of fish samples

The acid digesting technique was utilized to evaluate the heavy metal concentration in the gills, liver, and muscles of selected fish [28]. For the analysis, fish organs were isolated and digested using a mixture of analytical grade $HNO_3$ (65%—SIGMA-ALDRICH) and $HClO_4$ (60%—DAEJUNG) in a 3:1 volume ratio. The digestion process followed the Clesceri, Eaton [40] method, lasting 4–5 hours, and was carried out on a hot plate (ANEX Deluxe Hot Plate AG-2166-EX) at 100˚C. After digestion, the samples were allowed to cool to room temperature (25˚C) and then filtered through Whatman No. 42 filter paper. The filtered samples were then stored in clean, acid-washed polyethylene containers in a refrigerator at 4˚C for further analysis.

## 2.4 Detection of TTEs using Atomic Absorption Spectrophotometer

The stored samples were sent to the Central Laboratory of Muhammad Nawaz Shareef University of Agriculture (MNSUA) in Multan for the detection of TTEs (Cd, Cu, Pb, Ni). The concentrations of TTEs were determined using an acetylene air flame Spectroscopy using an Atomic Absorption Spectrophotometer (AAS) (Analytik Jena: NovAA 400 P). The samples were subsequently tested regarding the content of TTEs in accordance with the prescribed instrument settings, including a specific limit of detection (Table 1). A blank was generated for every sample, and modifications were made using a reference to the blank to ensure reproducibility and quality control of the analysis performed at Central Laboratory, MNSUA. The accuracy and precision of the tests were verified by comparing the results with the reference material (CRM IAEA 407) supplied by the International Atomic Energy Agency (IAEA). Additionally, the analysis of blanks and standards demonstrated satisfactory performance in

**Table 1. Condition of atomic absorption spectrometer used for the detection of heavy metal concentration.**

| Metals | Wavelength (nm) | Detection limits | Gas | Support |
|---|---|---|---|---|
| Cadmium (Cd) | 228.8 | 0.10 mg/kg | Acetylene | Air |
| Copper (Cu) | 248.33 | 0.05 mg/kg | Acetylene | Air |
| Lead (Pb) | 217 | 0.5 mg/kg | Acetylene | Air |
| Nickel (Ni) | 232 | 0.5 mg/kg | Acetylene | Air |

heavy metal determination, with recoveries ranging from 95% to 101% for the metals examined.

## 2.5 Human health risk assessment

Human health risk assessment of TTEs in fish muscles is important to highlight the bioaccumulation and harmful effects of trace elements like Cd, Cu, Pb, and Ni, which can pose serious health risks to consumers. Understanding the levels of TTEs in fish helps in identifying potential hazards, guiding regulatory limits, and ensuring food safety, thereby protecting public health from long-term exposure to toxic substances. The current study considered two main categories (regular and seasonal) of fish consumers. Regular consumers eat fish daily for the whole year. These are mostly fishermen or the population that lives very close to the rivers and have easy and cheap access to freshwater fish. Seasonal consumers eat fish only during cooler months and avoid eating fish during hot weather. These two categories were further divided into two more categories (mature and juvenile). Mature consumers eat a higher quantity of fish and have a higher body weight (kg), while juvenile consumers eat a lower portion of fish and have comparatively lower body weight. The human health risk assessment is carried out based on a few mathematical equations. However, these indices have certain assumptions like exposure pathways (THQ and TTHQ assume that all exposure to TTEs occurs via specified pathways like fish) [41], proportionality of dose and effect (These indices assume a linear relationship between the exposure dose and health risk) [42], cumulative effects (THQ measures risk from individual metals, while TTHQ accounts for the combined exposure from multiple metals. This assumes additive toxicity, but in reality, the metals could have synergistic or antagonistic interactions) [41, 42], susceptibility (It assumes that all individuals in the population are equally susceptible to metal toxicity, overlooking age, genetic factors, or pre-existing health conditions, which can influence individual vulnerability) [43], and steady-state exposure (These models assume constant exposure over a specific period, ignoring variations in metal concentrations or exposure frequency) [43]. Furthermore, there are a few limitations to the successful prediction of human health risk involved for example, THQ and TTHQ do not account for differences in the bioavailability of metals, and the hazard quotient method (used in both THQ and TTHQ) is based on reference doses (RfD), which are themselves estimates with inherent uncertainties. If these thresholds are set too conservatively, the assessment could overestimate risks [41, 42]. Moreover, MPI provides an aggregate measure of metal contamination in fish but does not directly reflect the toxicity or health risk posed by individual metals. Some metals, like cadmium, are more toxic to humans at lower concentrations than others (e.g., zinc), yet the MPI does not differentiate between their varying health risks.

These health indices used here have a few constants to estimate the risks associated with the consumption of fish having toxicity from selected TTEs (**Table 2**).

**2.5.1 Sensitivity to key input parameters.** Human health risk assessment indices (THQ, TTHQ, and MPI) are sensitive to certain key input parameters. Among these input parameters, THQ and TTHQ are inversely proportional to the RfD. Furthermore, Small increases in the ingestion rate can significantly raise the calculated THQ/TTHQ, especially for populations that consume large quantities of fish regularly (e.g., fishing communities). An increase in exposure frequency or duration will increase the THQ/TTHQ, indicating a higher risk of adverse health effects. THQ and TTHQ calculations account for body weight, which affects how contaminants are distributed and metabolized in the body. The THQ decreases with increasing body weight, meaning children or individuals with lower body weight are at greater risk per unit of exposure than adults. Incorrect estimates of average body weight could skew the risk assessment, particularly for vulnerable groups like children. The current study assumed these

**Table 2. Different terms with definitions and values are used for the health risk assessment of toxic trace elements in fish muscles.**

| Term used | Definition | Value |
|---|---|---|
| $IR_M$ | Ingestion rate of mature person | 0.20kg/day |
| $IR_J$ | Ingestion rate of juvenile person | 0.10kg/day |
| $EF_s$ | Exposure frequency | 150 days/year |
| $EF_r$ | Exposure frequency | 365 days/year |
| $ED_M$ | Exposure duration of mature person | 30 years |
| $ED_J$ | Exposure of juvenile person | 12 years |
| $BW_M$ | Body weight of mature person | 70 kg |
| $BW_J$ | Body weight of juvenile person | 30 kg |
| $AT_M$ | Average days (Exposure) of mature person | 10,950 days |
| $AT_J$ | Average days (Exposure) of juvenila person | 4380 days |

input parameters to be constant for a better assessment of human health risks associated with fish consumption.

**2.5.2 Ingestion exposure estimation.** The estimation of ingestion exposure ($^{in}Exp^R_M$) of selected TTEs in selected categories was calculated based on the following equations (**Eqs 1–4**) [2].

$$Exp^r_M = \frac{C \times IR_M \times IR_r \times ED_r}{BW_M \times AT_M} \qquad \text{Eq1}$$

$$Exp^r_J = \frac{C \times IR_J \times IR_r \times ED_r}{BW_J \times AT_J} \qquad \text{Eq2}$$

$$Exp^s_M = \frac{C \times IR_M \times IR_s \times ED_s}{BW_M \times AT_M} \qquad \text{Eq3}$$

$$Exp^s_J = \frac{C \times IR_J \times IR_s \times ED_s}{BW_J \times AT_J} \qquad \text{Eq4}$$

Where $Exp^r_M$ is the ingestion exposure of TTEs in mature regular consumers, $Exp^r_J$ is the ingestion exposure of TTEs in juvenile regular consumers, $Exp^s_M$ is the ingestion exposure of TTEs in mature seasonal consumers, and $Exp^s_J$ is the ingestion exposure of TTEs in juvenile seasonal consumers.

**2.5.3 Target hazardous quotients estimation.** Target hazardous quotients (THQs) are utilized to assess the potential non-carcinogenic health risks associated with exposure to TTEs present in the edible parts of fish (muscles) in accordance with the health risk assessment guidelines provided by the US EPA [44, 45]. The THQs due to fish consumption for the regular and seasonal (mature and juvenile) consumers were calculated using the following equations (**Eqs 5–8**).

$$THQ^r_M = \frac{Exp^r_M}{RfD} \qquad \text{Eq5}$$

$$THQ^r_J = \frac{Exp^r_J}{RfD} \qquad \text{Eq6}$$

$$THQ_M^s = \frac{Exp_M^s}{RfD} \qquad \text{Eq7}$$

$$THQ_J^s = \frac{Exp_J^s}{RfD} \qquad \text{Eq8}$$

THQ represents the Hazard Quotient, calculated based on ingestion at the corresponding exposure level; RfD represents the reference dose for the potentially hazardous health effects caused by contaminants through ingestion of TTEs. The current study used RfD value of 0.001 [46], 0.04 [47], 0.004 [48], and 0.02 [48] for Cd, Cu, Pb, and Ni, respectively. THQ > 1 represents the potentially toxic effects of TTEs on human consumption, while THQ < 1 represents that fish is safe for human consumption.

**2.5.4 Total target hazardous quotient estimation.** The Total target hazardous quotient (TTHQ) is the sum of THQs for individual TTE and represents the cumulative exposure effects of all TTEs on human health. Like THQ, the TTHQ > 1 represents the potentially toxic effects of TTEs on human consumption, while TTHQ < 1 represents that fish is safe for human consumption [2]. The TTHQs dues to fish consumption for the regular and seasonal (mature and juvenile) consumers were calculated using the following equations (**Eqs 9–12**).

$$TTHQ_M^r = \sum_i^n THQ_M^r \qquad \text{Eq9}$$

$$TTHQ_J^r = \sum_i^n THQ_J^r \qquad \text{Eq10}$$

$$TTHQ_M^s = \sum_i^n THQ_M^s \qquad \text{Eq11}$$

$$TTHQ_J^s = \sum_i^n THQ_J^s \qquad \text{Eq12}$$

**2.5.5 Metal pollution index (MPI).** The metal pollution index of TTEs (MPI$_{TTE}$) is determined using the following equation [49, 50]:

$$MPI_{TTE} = (M1 \times M2 \times \ldots \times Mn)^{1/n} \qquad \text{Eq13}$$

Where $M1$ is the concentration (mg/kg) of the first TTE, $M2$ is the concentration of the second TTE, and Mn is the concentration of the $nth$ TTE in the muscle of fish, while $n$ is the number of TTE studied.

## 2.6 Statistical analysis

The data collected in this research are presented as mean ± S.E. Normal distribution was observed within each group, and statistical analysis was conducted using one-way analysis of variance (ANOVA) with IBM SPSS Statistics software (version 20). Post hoc Duncan multiple range test with α < 0.05 was used to determine differences in mean values. Furthermore, the data were also subjected to Pearson's correlation analysis using Minitab (Version: 19.1.1)) to determine the correlation between various fish species for the accumulation of TTEs at the right and left banks of Punjnad headworks. We also applied Principal component analysis (PCA) and hierarchical cluster analysis (HCA) by using PAST (version: 4.03) to explore the relationship between fish species and studied TTEs.

## 3. Results

### 3.1 Physicochemical parameters of the water from study sites

The physicochemical parameters of water quality of two study sites (RB and LB) revealed variations in several key parameters. The temperature changes significantly between seasons, from 21.32 ± 0.99 in RB during winter to 27.24 ± 0.98 in LB during summer. Electrical conductivity also decreased from 401.66 ± 5.30 μS/cm in RB during winter to 257.73 ± 18.07 μS/cm in LB during summer, suggesting changes in the water's ability to conduct electricity as the water temperature rises. Dissolved oxygen (DO) levels, crucial for aquatic life, significantly decreased from 4.14 ± 0.21 mg/L in LB during winter to 0.79 ± 0.13 mg/L in RB during spring. Total dissolved solids (TDS) also showed a significant change, with values of 177.22 ± 8.89 mg/L in RB during spring and 202.09 ± 4.50 mg/L in RB during winter. The pH exhibited a significant decline from 8.13±0.37 in LB during winter to 7.13± 0.44 in RB in summer, indicating a shift towards an acidic nature as temperature rises (Table 3).

### 3.2 Accumulation of toxic trace elements (TTEs) in different organs of fish

A study was conducted to assess the accumulation of four TTEs (Cd, Cu, Pb, and Ni) in three different organs (liver, gills, and muscles) in five fish species (*R. rita*, *S. sarwari*, *W. attu*, *M. armatus*, and *C. mrigala*) during three different seasons (winter, spring, and summer) from the right (RB) and the left banks.

**3.2.1 Accumulation of toxic trace elements (TTEs) during winter.** The season-wise analysis of accumulation (mg/kg) of TTEs showed that during winter, across all species and TTEs, the liver consistently showed the highest concentration of TTEs, followed by gills and muscles. This trend is expected because the liver is a primary detoxifying organ, where metals accumulate more prominently. *Wallago attu* consistently showed the highest accumulation among all metals and organs compared to other species, particularly in the liver, where it has the highest recorded values for Cd (10.12±2.18 in LB and 15.25±2.47 in RB), Cu (5.65±1.57 in LB and 7.48±1.81 in RB), Pb (41.61±4.99 in LB and 46.03±10.56 in RB), and Ni (8.94±2.21 in LB and 11.68±1.23 in RB). On the other hand, *C. mrigala* has the lowest accumulation TTEs having the significantly lower accumulation (Cd (1.38±0.15 in LB and 2.08±0.52 in RB), Cu (0.57±0.06 in LB and 1.64±0.21 in RB), Pb (8.29±1.65 in LB and 11.87±1.90 in RB), and Ni (1.48±0.24 in LB and 2.39±0.26 in RB) in fish muscles. Furthermore, the pattern of TTEs' accumulation in fish species was in the order of *W. attu* > *M. armatus* > *S. sarwari* > *R. rita* > *C. mrigala* for all TTEs and fish organs except for the Cu accumulation in the RB, where the

**Table 3. Physicochemical parameters of the water samples collected from the left and right banks of Punjnad headworks during the three seasons.**

| Physicochemical parameters | Study Site | Season | | |
|---|---|---|---|---|
| | | Winter | Spring | Summer |
| Water Temperature (˚C) | Left Bank | 21.41 ± 0.95 | 23.07 ± 0.76 | 27.24 ± 0.98 |
| | Right Bank | 21.32 ± 0.99 | 24.28 ± 1.22 | 26.60 ± 0.65 |
| Electrical conductivity (μS/cm) | Left Bank | 386.28 ± 3.58 | 374.12 ± 16.85 | 257.73 ± 18.07 |
| | Right Bank | 401.66 ± 5.30 | 352.27 ± 11.61 | 261.55 ± 7.57 |
| Dissolved oxygen (mg/L) | Left Bank | 4.14 ± 0.21 | 2.84 ± 0.49 | 2.08 ± 0.18 |
| | Right Bank | 3.84 ± 0.50 | 0.79 ± 0.13 | 2.05 ± 0.09 |
| Total dissolved solid (mg/L) | Left Bank | 196.34 ± 3.50 | 191.09 ± 6.98 | 185.08 ± 6.06 |
| | Right Bank | 202.09 ± 4.50 | 177.22 ± 8.89 | 192.19 ± 4.11 |
| pH | Left Bank | 8.13 ± 0.37 | 7.60 ± 0.26 | 7.35 ± 0.48 |
| | Right Bank | 7.92± 0.41 | 7.54 ± 0.32 | 7.13± 0.44 |

trend was *M. armatus* > *W. attu* > *S. sarwari* > *R. rita* > *C. mrigala*. Generally, the RB of the river shows higher concentrations of all metals compared to the left bank for all species and tissues, indicating a potential difference in pollution levels or water flow patterns between the two banks. Seasonal or anthropogenic fluctuations in water levels, such as through dams or irrigation systems, can redistribute pollutants [51]. The difference is most pronounced in *W. attu*, where the increase in accumulation from the LB to RB is highly significant, especially for lead (Pb) in the liver (41.61±14.99 in LB vs. 46.03±10.56 in RB) during winter (**Table 4**).

**3.2.2 Accumulation of toxic trace elements (TTEs) during spring.** Similar to the winter, the spring season also indicated that among all species and TTEs, the liver consistently showed the highest concentration of TTEs, followed by gills and muscles. *Wallago attu* consistently showed the highest accumulation among all metals and organs compared to other species with significantly higher accumulation in the liver, with the value of 9.37±0.48 in LB and 13.92 ±0.87 in RB for Cd, 11.21±1.30 in LB and 12.36±0.47 in RB for Cu, 45.04±2.15 in LB for Pb, and 13.33±0.99 in LB and 17.92±1.46 in RB for Ni, except for the Pb (46.01±2.19) in RB which was higher in *M. armatus*. On the other hand, *C. mrigala* showed significantly lower accumulation (Cd (4.16±0.83 in RB), Cu (2.32±0.27 in LB and 3.55±0.54 in RB), Pb (11.19±0.89 in LB and 12.65±1.16 in RB), and Ni (3.59±0.40 in LB and 5.21±0.41 in RB)) in fish muscles except for the Cd concentration (5.02±0.70) in the LB which was found to be lowest in the *R. rita*. Furthermore, the pattern of TTEs' accumulation in fish species was mostly in the order of *W.*

**Table 4. Accumulation (mg/kg) of toxic trace elements (TTEs) in different organs of five fish species collected from the right (RB) and left banks (LB) of Punjnad headworks, Uch Sharif during winter.**

| Left Bank (LB) of Punjand river | | | | | | | | | | | | |
|---|---|---|---|---|---|---|---|---|---|---|---|---|
| TTE | Cadmium (Cd) | | | Copper (Cu) | | | Lead (Pb) | | | Nickel (Ni) | | |
| Species | Gills | Liver | Muscles | Gills | Liver | Muscles | Gills | Liver | Muscles | Gills | Liver | Muscles |
| *R. rita* | 4.86±0.55[B] | 6.80±0.78[A] | 2.46 ±0.44[C] | 1.80±0.20[B] | 3.72 ±0.51[A] | 1.21 ±0.13[B] | 13.90 ±1.84[B] | 27.66±3.75[A] | 11.96 ±1.32[C] | 3.75 ±0.64[AB] | 4.67±1.10[A] | 2.68 ±0.36[B] |
| *S. sarwari* | 5.07±0.72[B] | 7.86±1.36[A] | 2.88 ±0.53[C] | 2.42±0.63[B] | 4.36 ±1.15[A] | 2.12 ±0.31[B] | 16.88 ±1.94[B] | 31.64±7.65[A] | 13.70 ±3.42[C] | 4.83 ±1.28[AB] | 6.46±1.39[A] | 3.88 ±0.58[B] |
| *W. attu* | 6.74±1.42[B] | 10.12 ±2.18[A] | 4.14 ±0.60[C] | 3.91±0.74[AB] | 5.65 ±1.57[A] | 3.09 ±0.54[B] | 27.26 ±5.22[B] | 41.61±4.99[A] | 14.26 ±2.01[C] | 7.50 ±1.86[AB] | 8.94±2.21[A] | 4.99 ±1.13[B] |
| *M. armatus* | 5.47±1.49[A] | 8.37±1.69[A] | 3.58 ±0.40[C] | 2.89±0.81[B] | 4.94 ±0.55[A] | 2.58 ±0.48[B] | 21.26 ±2.65[B] | 38.38±6.37[A] | 13.83 ±2.52[C] | 6.03 ±0.96[AB] | 7.55±1.54[A] | 4.44 ±0.59[B] |
| *C. mrigala* | 3.42±0.35[B] | 6.13±1.14[A] | 1.45 ±0.30[C] | 1.38±0.15[B] | 2.89 ±0.64[A] | 0.57 ±0.06[C] | 11.75 ±1.89[B] | 19.29±2.63[A] | 8.29±1.65[C] | 3.24±0.65[A] | 3.72±0.64[A] | 1.48 ±0.24[B] |
| Right Bank (RB) of Punjand river | | | | | | | | | | | | |
| TTE | Cadmium (Cd) | | | Copper (Cu) | | | Lead (Pb) | | | Nickel (Ni) | | |
| Species | Gills | Liver | Muscles | Gills | Liver | Muscles | Gills | Liver | Muscles | Gills | Liver | Muscles |
| *R. rita* | 8.25±1.57[B] | 10.64 ±1.25[A] | 4.09±0.5[C] | 2.04±0.35[B] | 4.89 ±1.12[A] | 1.79 ±0.28[C] | 18.65 ±3.58[B] | 29.88±8.00[A] | 15.25 ±1.92[C] | 4.77±0.61[B] | 7.76±1.33[A] | 3.01 ±0.40[C] |
| *S. sarwari* | 9.34±1.03[B] | 13.82 ±1.94[A] | 5.23 ±0.63[C] | 4.38±1.01[A] | 5.28 ±0.74[A] | 2.68 ±0.38[B] | 20.63 ±3.49[B] | 36.25±8.27[A] | 15.73 ±3.75[C] | 5.85 ±0.83[AB] | 8.28±1.97[A] | 4.96 ±0.55[B] |
| *W. attu* | 12.04 ±2.18[B] | 15.25 ±2.47[A] | 8.17 ±1.90[C] | 5.67±1.10[A] | 7.48 ±1.81[A] | 4.90 ±0.94[A] | 22.53 ±3.81[B] | 46.03 ±10.56[A] | 20.36 ±2.48[C] | 8.07±1.51[B] | 11.68 ±1.23[A] | 6.30 ±0.76[B] |
| *M. armatus* | 9.52±1.12[B] | 14.00 ±1.82[A] | 6.54 ±1.06[C] | 4.95±1.03[A] | 6.10 ±0.85[A] | 2.87 ±0.80[B] | 22.77 ±4.46[B] | 40.75±6.89[A] | 15.83 ±3.35[C] | 6.66±0.67[B] | 10.05 ±1.41[A] | 5.10 ±1.45[B] |
| *C. mrigala* | 5.95±1.59[B] | 9.15±1.83[A] | 2.08 ±0.52[C] | 2.00±0.34[B] | 4.33 ±0.79[A] | 1.64 ±0.21[B] | 14.97 ±2.83[B] | 24.61±5.87[A] | 11.87 ±1.90[C] | 3.74±0.50[B] | 6.82±1.12[A] | 2.39 ±0.26[B] |

†Different letters (A-C) in the same row and same TTE show significant differences in its accumulation among different fish organs based on the Duncan multiple range test (α < 0.05).

**Table 5. Accumulation (mg/kg) of toxic trace elements (TTEs) in different organs of five fish species collected from the right (RB) and left banks (LB) of Punjnad headworks, Uch Sharif during spring.**

| | Left Bank (LB) of Punjand river | | | | | | | | | | | |
|---|---|---|---|---|---|---|---|---|---|---|---|---|
| TTE | Cadmium (Cd) | | | Copper (Cu) | | | Lead (Pb) | | | Nickel (Ni) | | |
| Species | Gills | Liver | Muscles | Gills | Liver | Muscles | Gills | Liver | Muscles | Gills | Liver | Muscles |
| *R.rita* | 6.80±0.65[B] | 10.79 ±0.81[A] | 5.02±0.70[C] | 4.75±0.30[B] | 6.52±0.70[A] | 3.50 ±0.63[C] | 23.44 ±2.42[B] | 32.29 ±3.20[A] | 14.06 ±2.69[C] | 6.62±0.69[B] | 9.89±0.60[A] | 5.24 ±0.52[C] |
| *S. sarwari* | 7.95±0.54[B] | 12.31 ±0.94[A] | 5.78±0.62[C] | 6.50±1.34[B] | 8.78±0.75[A] | 4.26 ±0.58[C] | 22.94 ±1.43[B] | 35.10 ±1.89[A] | 15.54 ±3.02[C] | 6.94±0.55[B] | 10.26 ±1.53[A] | 5.14 ±0.80[B] |
| *W. attu* | 9.37±0.48[B] | 14.65 ±1.25[A] | 6.65±0.53[C] | 8.82±0.74[B] | 11.21 ±1.30[A] | 6.22 ±0.56[C] | 33.56 ±1.71[B] | 45.04 ±2.15[A] | 18.78 ±4.70[C] | 9.12±0.33[B] | 13.33 ±0.99[A] | 6.65 ±0.66[C] |
| *M. armatus* | 8.43±1.01[B] | 13.60 ±1.83[A] | 6.15±0.82[B] | 7.54±0.91[B] | 10.74 ±0.50[A] | 4.86 ±0.71[C] | 25.28 ±2.10[B] | 38.22 ±3.10[A] | 15.72 ±0.93[C] | 8.50±1.04[B] | 12.09 ±0.57[A] | 6.73 ±1.06[B] |
| *C. mrigala* | 7.07±0.31[B] | 10.05 ±0.83[A] | 5.31±0.37[C] | 4.51±0.60[B] | 6.82±0.41[A] | 2.32 ±0.27[C] | 14.71 ±1.43[B] | 24.42 ±1.81[A] | 11.19 ±0.89[C] | 6.27±0.62[B] | 8.32±0.95[A] | 3.59 ±0.40[C] |
| | Right Bank (RB) of Punjand river | | | | | | | | | | | |
| TTE | Cadmium (Cd) | | | Copper (Cu) | | | Lead (Pb) | | | Nickel (Ni) | | |
| Species | Gills | Liver | Muscles | Gills | Liver | Muscles | Gills | Liver | Muscles | Gills | Liver | Muscles |
| *R.rita* | 11.76 ±1.04[B] | 15.54 ±0.98[A] | 6.43±0.90[C] | 5.99±0.16[B] | 8.75±0.73[A] | 3.98 ±0.65[C] | 20.80 ±1.60[B] | 36.89 ±3.44[A] | 16.12 ±1.15[C] | 8.41±1.18[B] | 12.02 ±1.05[A] | 6.10 ±0.81[C] |
| *S. sarwari* | 13.54 ±0.79[B] | 18.03 ±1.42[A] | 7.42±0.97[C] | 8.01±0.74[B] | 9.58±0.51[A] | 5.31 ±0.51[C] | 26.89 ±1.29[B] | 37.61 ±1.78[A] | 17.91 ±1.23[C] | 9.79±0.32[B] | 13.05 ±1.10[A] | 6.74 ±0.70[C] |
| *W. attu* | 13.92 ±0.87[B] | 20.59 ±2.10[A] | 10.06 ±1.25[C] | 10.16 ±0.64[B] | 12.36 ±0.47[A] | 7.09 ±0.65[C] | 23.81 ±1.38[B] | 42.46 ±2.66[A] | 17.50 ±1.09[C] | 13.07 ±0.51[B] | 17.92 ±1.46[A] | 7.06 ±0.43[C] |
| *M. armatus* | 12.73 ±0.74[B] | 19.78 ±0.97[A] | 7.86±0.65[C] | 8.97±0.56[B] | 11.01 ±0.69[A] | 5.92 ±0.48[C] | 22.57 ±0.83[B] | 46.01 ±2.19[A] | 17.83 ±1.29[C] | 10.87 ±0.58[B] | 15.33 ±1.07[A] | 6.90 ±0.85[C] |
| *C. mrigala* | 8.99±0.57[B] | 12.26 ±0.74[A] | 4.16±0.83[C] | 5.90±0.74[B] | 8.74±0.46[A] | 3.55 ±0.54[C] | 18.66 ±1.94[B] | 29.69 ±3.40[A] | 12.65 ±1.16[C] | 7.92±0.45[B] | 11.87 ±0.96[A] | 5.21 ±0.41[C] |

†Different letters (A-C) in the same row and same TTE show significant differences in its accumulation among different fish organs based on the Duncan multiple range test ($\alpha < 0.05$).

*attu > M. armatus > S. sarwari > R. rita > C. mrigala* for all TTEs and fish organs in both LB and RB except Cd accumulation in gills and muscles and Cu accumulation in liver which was in order of *W. attu > M. armatus > S. sarwari > C. mrigala > R. rita* and Ni accumulation in muscles with the order of (*M. armatus > W. attu > R. rita >S. sarwari > C. mrigala*) in the LB and gills and muscles accumulation in the Cd and Pb respectively from the RB during summer. It is observed that the RB of the river shows higher concentrations of all metals compared to the left bank for all species and tissues except for the Pb concentration in the gills of *R. rita*, *W. attu*, and *M. armatus* as well as in the liver and muscles of *W. attu*. It indicates a potential difference in pollution levels or water flow patterns between the two banks. The difference is most pronounced in *M. armatus*, where the increase in accumulation from the LB to RB is highly significant, especially for lead (Pb) in the liver (38.22±3.10 in LB vs. 46.01±2.19 in RB) during spring (Table 5).

**3.2.3 Accumulation of toxic trace elements (TTEs) during summer.** The accumulation of TTEs during summer had a mixed trend in various organs and fish species. *Wallago attu* mostly showed the highest accumulation among all metals and organs compared to other species with significantly higher accumulation in the liver, with the value of 23.32±0.56 in LB and 26.42±0.97 in RB for Cd, 20.72±0.42 in LB for Cu, 47.76±1.44 in LB for Pb, and 22.05±0.76 in LB and 21.94±1.55 in RB for Ni, except Cu accumulation (19.99±0.35) in the RB which was higher in *S. sarwari* and Pb concentration (53.14±2.35) in the RB which was highest in *M*

**Table 6. Accumulation (mg/kg) of toxic trace elements (TTEs) in different organs of five fish species collected from the right (RB) and left banks (LB) of Punjnad headworks, Uch Sharif during summer.**

| | Left Bank (LB) of Punjand river | | | | | | | | | | | |
|---|---|---|---|---|---|---|---|---|---|---|---|---|
| **TTE** | **Cadmium (Cd)** | | | **Copper (Cu)** | | | **Lead (Pb)** | | | **Nickel (Ni)** | | |
| **Species** | Gills | Liver | Muscles | Gills | Liver | Muscles | Gills | Liver | Muscles | Gills | Liver | Muscles |
| *R. rita* | 11.43 ±0.55[B] | 15.04 ±0.48[A] | 7.47±1.10[C] | 7.77 ±0.67[B] | 11.63 ±1.00[A] | 5.98±0.45[C] | 23.60 ±1.25[B] | 36.44 ±1.35[A] | 16.88 ±1.72[C] | 10.76 ±0.91[B] | 12.80 ±0.38[A] | 7.74±0.75[C] |
| *S. sarwari* | 14.58 ±2.34[A] | 14.80 ±3.25[A] | 10.15 ±0.82[A] | 12.22 ±1.01[B] | 15.32 ±0.90[A] | 9.29±1.27[B] | 26.18 ±0.27[B] | 36.87 ±1.11[A] | 20.91 ±6.52[C] | 11.88 ±0.99[B] | 16.56 ±1.03[A] | 10.41 ±1.06[C] |
| *W. attu* | 17.46 ±1.87[B] | 23.32 ±0.56[A] | 12.96 ±1.18[C] | 16.69 ±2.01[B] | 20.72 ±0.42[A] | 14.33±0.38 | 32.60 ±1.78[B] | 47.76 ±1.44[A] | 22.99 ±2.64[C] | 17.58 ±1.25[B] | 22.05 ±0.76[A] | 15.45 ±0.12[C] |
| *M. armatus* | 14.94 ±0.21[B] | 20.58 ±0.90[A] | 12.36 ±1.35[C] | 12.90 ±1.20[B] | 18.22 ±0.54[A] | 10.49 ±0.34[C] | 28.94 ±1.17[B] | 46.60 ±1.52[A] | 21.80 ±1.43[C] | 12.71 ±0.36[B] | 17.36 ±0.91[A] | 10.69 ±1.01[C] |
| *C. mrigala* | 8.10±1.00[B] | 13.61 ±1.37[A] | 5.01±1.45[C] | 8.09 ±0.37[B] | 10.67 ±0.76[A] | 4.76±0.96[C] | 15.91 ±1.54[B] | 25.52 ±2.15[A] | 11.30 ±0.83[C] | 10.00 ±1.62[B] | 14.17 ±1.19[A] | 4.90±0.94[C] |
| | Right Bank (RB) of Punjand river | | | | | | | | | | | |
| **TTE** | **Cadmium (Cd)** | | | **Copper (Cu)** | | | **Lead (Pb)** | | | **Nickel (Ni)** | | |
| **Species** | Gills | Liver | Muscles | Gills | Liver | Muscles | Gills | Liver | Muscles | Gills | Liver | Muscles |
| *R. rita* | 15.65 ±1.38[B] | 19.91 ±1.36[A] | 10.73 ±0.79[C] | 12.72 ±1.51[B] | 15.42 ±0.72[A] | 7.7±0.48[C] | 27.29 ±0.92[B] | 37.09 ±2.19[A] | 19.37 ±0.62[C] | 12.52 ±1.38[B] | 17.63 ±0.76[A] | 9.20±1.47[C] |
| *S. sarwari* | 17.19 ±0.32[B] | 23.79 ±0.30[A] | 13.43 ±1.70[C] | 15.42 ±1.47[B] | 19.99 ±0.35[A] | 10.87 ±0.37[C] | 30.74 ±1.81[B] | 45.67 ±1.03[A] | 24.17 ±1.29[C] | 14.33 ±1.46[B] | 19.76 ±1.67[A] | 12.09 ±0.57[C] |
| *W. attu* | 22.40 ±1.22[B] | 26.42 ±0.97[A] | 13.96 ±1.23[C] | 13.48 ±1.22[B] | 17.72 ±0.45[A] | 10.79 ±0.92[C] | 34.65 ±1.16[B] | 52.02 ±2.31[A] | 29.94 ±7.37[B] | 17.39 ±1.14[B] | 21.94 ±1.55[A] | 14.20 ±1.28[C] |
| *M. armatus* | 17.33 ±1.14[B] | 22.14 ±1.33[A] | 13.39 ±0.91[C] | 14.97 ±0.44[B] | 18.07 ±0.56[A] | 11.34 ±1.19[C] | 28.30 ±0.97[B] | 53.14 ±2.35[A] | 22.67 ±2.61[C] | 18.64 ±0.65[B] | 21.53 ±0.88[A] | 12.75 ±0.84[C] |
| *C. mrigala* | 12.16 ±0.85[B] | 16.46 ±1.50[A] | 7.56±1.01[C] | 8.64 ±0.38[B] | 12.81 ±1.25[A] | 5.67±0.51[C] | 23.76 ±0.73[B] | 37.17 ±3.16[A] | 11.80 ±1.47[C] | 9.32 ±0.87[B] | 12.88 ±1.15[A] | 5.23±1.47[C] |

†Different letters (A-C) in the same row and the same TTE show significant differences in its accumulation among different fish organs based on the Duncan multiple range test ($\alpha < 0.05$)

*armatus. C. mrigala* has the lowest accumulation of TTEs with value of 5.01±1.45 in LB and 7.56±1.0 for Cd, 4.76±0.96 in LB and 5.67±0.51 in RB for Cu, 11.30±0.83 in LB and 11.80±1.47 in RB for Pb, and 4.90±0.94 in LB and 5.23±1.47 in RB for Ni in fish muscles. Moreover, the pattern of TTEs' accumulation in studied fish was mostly in the order of *W. attu > M. armatus > S. sarwari > R. rita > C. mrigala* for most of the TTEs and fish organs in LB with a few exceptions, but it showed a mixed trend in RB during summer. It is observed that the RB of the river shows higher concentrations of all metals compared to the left bank for all species and tissues except for the Cu and Ni concentration in gills, liver, and muscles of *W. attu* and Cu and Pb concentration in the gills of *M. armatus*. The difference in concentration of TTEs between the banks is most pronounced in *C. mrigala*, where the increase in accumulation from the LB to RB is highly significant, especially for lead (Pb) in the liver (25.52±2.15 in LB vs. 37.17±3.16 in RB) during summer (**Table 6**).

## 3.3 Correlation analysis among fish species

In order to explore the relationship between fish species and the accumulation of TTEs from LB and RB, a correlation analysis was carried out. The results showed that across all metals and banks, the correlation coefficients between different fish species are generally high (above 0.90 in most cases), indicating strong positive relationships in metal accumulation. This suggests that if one species has high metal accumulation, the others are likely to show similarly high

**Table 7. Correlation table of TTEs' accumulation in studied fish species at left (LB) and right banks (RB) of Punjand headworks, Uch Sharif.**

| Cadmium (Cd)—Left bank (LB) | | | | | Cadmium (Cd)—Right bank (RB) | | | | |
|---|---|---|---|---|---|---|---|---|---|
| Fish Species | *R. rita* | *S. sarwari* | *W. attu* | *M. armatus* | Fish Species | *R. rita* | *S. sarwari* | *W. attu* | *M. armatus* |
| *S. sarwari* | 0.926 | | | | *S. sarwari* | 0.973 | | | |
| *W. attu* | 0.972 | 0.928 | | | *W. attu* | 0.96 | 0.958 | | |
| *M. armatus* | 0.972 | 0.914 | 0.98 | | *M. armatus* | 0.963 | 0.975 | 0.955 | |
| *C. mrigala* | 0.932 | 0.846 | 0.902 | 0.889 | *C. mrigala* | 0.973 | 0.986 | 0.962 | 0.969 |
| Copper (Cu)—Left bank (LB) | | | | | Copper (Cu)—Right bank (RB) | | | | |
| Fish Species | *R. rita* | *S. sarwari* | *W. attu* | *M. armatus* | Fish Species | *R. rita* | *S. sarwari* | *W. attu* | *M. armatus* |
| *S. sarwari* | 0.964 | | | | *S. sarwari* | 0.968 | | | |
| *W. attu* | 0.947 | 0.986 | | | *W. attu* | 0.958 | 0.956 | | |
| *M. armatus* | 0.977 | 0.98 | 0.977 | | *M. armatus* | 0.975 | 0.983 | 0.973 | |
| *C. mrigala* | 0.953 | 0.951 | 0.94 | 0.971 | *C. mrigala* | 0.955 | 0.93 | 0.971 | 0.945 |
| Lead (Pb)—Left bank (LB) | | | | | Lead (Pb)—Right bank (RB) | | | | |
| Fish Species | *R. rita* | *S. sarwari* | *W. attu* | *M. armatus* | Fish Species | *R. rita* | *S. sarwari* | *W. attu* | *M. armatus* |
| *S. sarwari* | 0.934 | | | | *S. sarwari* | 0.952 | | | |
| *W. attu* | 0.939 | 0.932 | | | *W. attu* | 0.92 | 0.951 | | |
| *M. armatus* | 0.938 | 0.945 | 0.956 | | *M. armatus* | 0.954 | 0.96 | 0.943 | |
| *C. mrigala* | 0.951 | 0.935 | 0.947 | 0.942 | *C. mrigala* | 0.944 | 0.947 | 0.876 | 0.934 |
| Nickel (Ni)—Left bank (LB) | | | | | Nickel (Ni)—Right bank (RB) | | | | |
| Fish Species | *R. rita* | *S. sarwari* | *W. attu* | *M. armatus* | Fish Species | *R. rita* | *S. sarwari* | *W. attu* | *M. armatus* |
| *S. sarwari* | 0.934 | | | | *S. sarwari* | 0.959 | | | |
| *W. attu* | 0.935 | 0.978 | | | *W. attu* | 0.949 | 0.951 | | |
| *M. armatus* | 0.968 | 0.959 | 0.962 | | *M. armatus* | 0.96 | 0.965 | 0.969 | |
| *C. mrigala* | 0.943 | 0.896 | 0.889 | 0.932 | *C. mrigala* | 0.907 | 0.862 | 0.903 | 0.877 |

levels. The correlations are consistently high across both banks, but the right bank (RB) shows slightly higher correlations in some cases, particularly for Cadmium (Cd) and Copper (Cu). This may indicate more uniform environmental conditions on the right bank that affect all species similarly. It is more evident that the correlation for Cadmium (Cd) between *S. sarwari* and *C. mrigala*, increased from 0.846 on the left bank to 0.986 on the right bank, showing a stronger relationship on the right bank. *Wallago attu* generally exhibited the highest correlations with other species, especially for Nickel (Ni), where it shows a highly significant correlation (0.978) with *S. sarwari* on the left bank and strong correlations with other species on both banks. At the same time, *C. mrigala* showed slightly lower correlations compared to other species, particularly on the left bank, which might indicate a different bioaccumulation pattern or ecological niche (**Table 7**). The correlation analysis reveals that the accumulation of toxic trace elements in fish species from the Punjand headworks is highly interrelated, with particularly strong correlations observed on the right bank. This suggests that environmental factors influencing metal accumulation are relatively uniform across species, especially on the right bank. The consistently high correlations underline the importance of monitoring multiple species to understand the broader environmental impact of heavy metal contamination in this area.

## 3.4 Comparison of TTEs' accumulation between RB and LB

The difference in the concentration of TTEs between the LB and RB of the river was visualized using a box plot. The results indicated that among all species, lead (Pb) showed the highest accumulation compared to other toxic trace elements (Cd, Cu, Ni), especially in the RB samples. This is evident from the box plots where Pb's range and median values significantly

exceed those of the other elements or most species. Moreover, the RB of the Punjand River generally exhibits higher bioaccumulation levels across all metals. This trend is particularly noticeable in *W. attu* and *S. sarwari*, where the difference is marked, indicating a higher contamination level on the right bank. Although variations exist, the general pattern of higher accumulation on the right bank is consistent across species, reinforcing the environmental impact on this side of the river. Among various fish species, *W. attu* showed the most significant accumulation for all elements, particularly for Pb and Ni, indicating that this species is highly susceptible to heavy metal bioaccumulation while, *C. mrigala* and *R. rita* displayed lower bioaccumulation levels, suggesting either a different habitat preference or varying metabolic rates in processing these elements (**Fig 2**). The box plots analysis illustrated a clear trend of higher heavy metal bioaccumulation in fish species from the right bank of the Punjand River, with Pb being the most prominent contaminant.

## 3.5 Seasonal comparison of TTEs' accumulation in fish

The season-wise comparison of TTEs' accumulation in different fish at LB of Punjand headworks showed a significant difference in the accumulation of TTEs across three seasons. The results showed that during summer, maximum accumulation of all trace elements in all species was observed. *W. attu* showed the highest accumulation of Pb during summer, while *C. mrigala* showed the lowest concentration of Cu during winter. Overall, the trend of accumulation of TTEs increased gradually from winter to summer, indicating a direct relationship between the accumulation of TTEs and the temperature of the aquatic ecosystem (**Fig 3**). Similar to the LB, RB also presented an increasing trend of TTEs accumulation from winter to summer. The maximum accumulation of TTEs was observed in *W. attu*, with the maximum concentration of Pb during summer. The winter season showed an overall low concentration of all TTEs across all studied fish. The lowest concentration of TTEs was determined in *C. mrigala* during winter for Cu concentration. Spring usually showed moderate accumulation of TTEs in all fish species (**Fig 4**).

## 3.6 Hierarchical cluster analysis (HCA) and principal component analysis (PCA)

The current study applied the Hierarchical cluster analysis (HCA) with the Wards linkage method and Euclidean distance as parameters to measure the similarity/variation for accumulation of TTEs in fish. HCA of fish species and the TTEs formed three clusters (group—1, group– 2, group– 3). Among fish species, cluster one (group—1) is formed due to the high content of TTEs in *W. attu*–LB, *W. attu*—RB, *S, sarwari*–RB, and *M. armatus*—RB. Cluster two (group—2) explained the moderate concentration of TTEs in *M. armatus*–LB, *S. sarwari*–LB, and *R. rita*–RB during the sampling seasons. Cluster three (group—3) is formed to indicate the lowest accumulation of TTEs in *R. rita*–LB, *C. mrigala*–RB, and *C. mrigala*–LB (**Fig 5A**). For the TTEs, Cluster 1 (group– 1) showed the lowest concentration of Cd–LB, Ni–LB, Cu–LB, and Cu–RB in fish at the Punjnad river. Cluster 2 (group– 2) is formed due to the moderate accumulation of Ni–RB and Cd–Rb at the study site. The formation of three different groups showed that Pb concentration was significantly higher at LB and RB of the river forming group– 3 (**Fig 5B**).

The study also carried out the principal component analysis (PCA) to demonstrate the similarity/variation in the distribution and behavior of TTEs and studied fish species. The results showed two principal components (PCs) with Eigenvalue $> 1$ and two PCs with Eigenvalue $< 1$. The Eigenvalues values of PC1, PC2, PC3, and PC4 were found to be 42.85, 1.57, 0.24, and 0.08, with the variance of 95.75, 3.51, 0.54, and 0.20%, respectively. It is

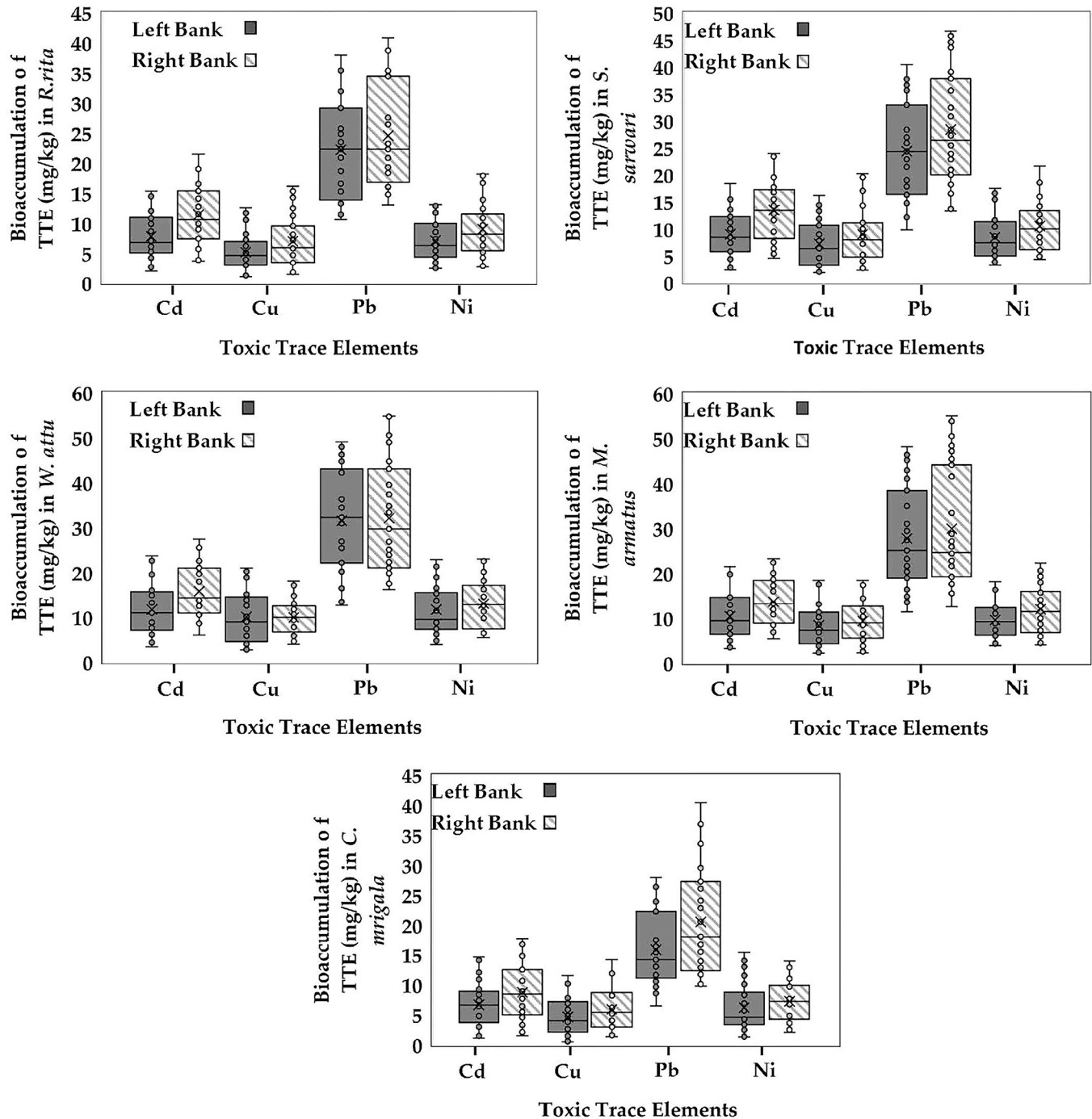

**Fig 2. Comparison of accumulation of four toxic trace elements (TTEs) (Cd, Cu, Pb, and Ni) in five fish species between left and right banks of Punjand headworks.**

observed that PC1 was positively dominated by Pb, having a loading value of 0.79, Cd dominated PC2, having a loading value of 0.85, PC3 was dominated by Cu, having a loading value of 0.73; a negatively dominated Ni with a loading value of -0.76 was indicated for PC4 (**Fig 5C**).

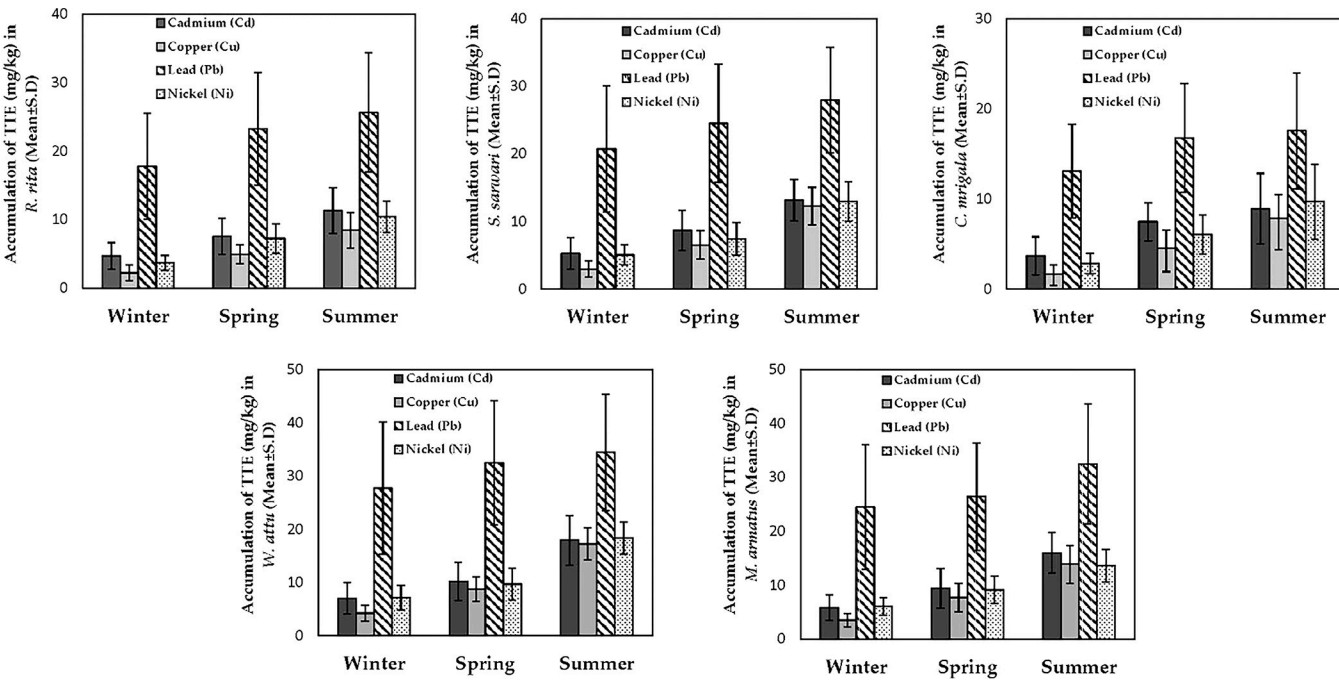

**Fig 3. Accumulation comparison of four toxic trace elements (TTEs) (Cd, Cu, Ni, and PB) in five fish species (*R. rita*, *S. sarwari*, *W. attu*, *M. armatus*, and *C. mrigala*) between three different sampling seasons (winter, spring, and summer) at the left bank (LB) of Punjnad headworks.**

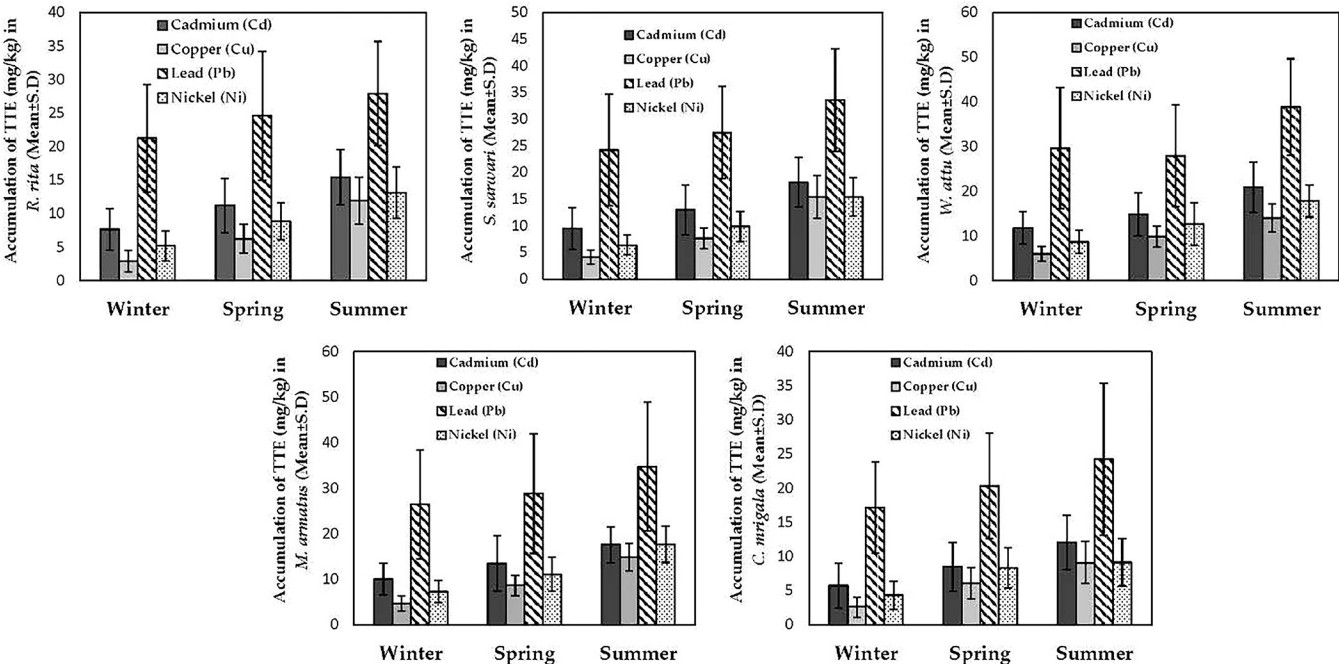

**Fig 4. Accumulation comparison of four toxic trace elements (TTEs) (Cd, Cu, Ni, and PB) in five fish species (*R. rita*, *S. sarwari*, *W. attu*, *M. armatus*, and *C. mrigala*) between three different sampling seasons (winter, spring, and summer) at the right bank (RB) of Punjnad headworks.**

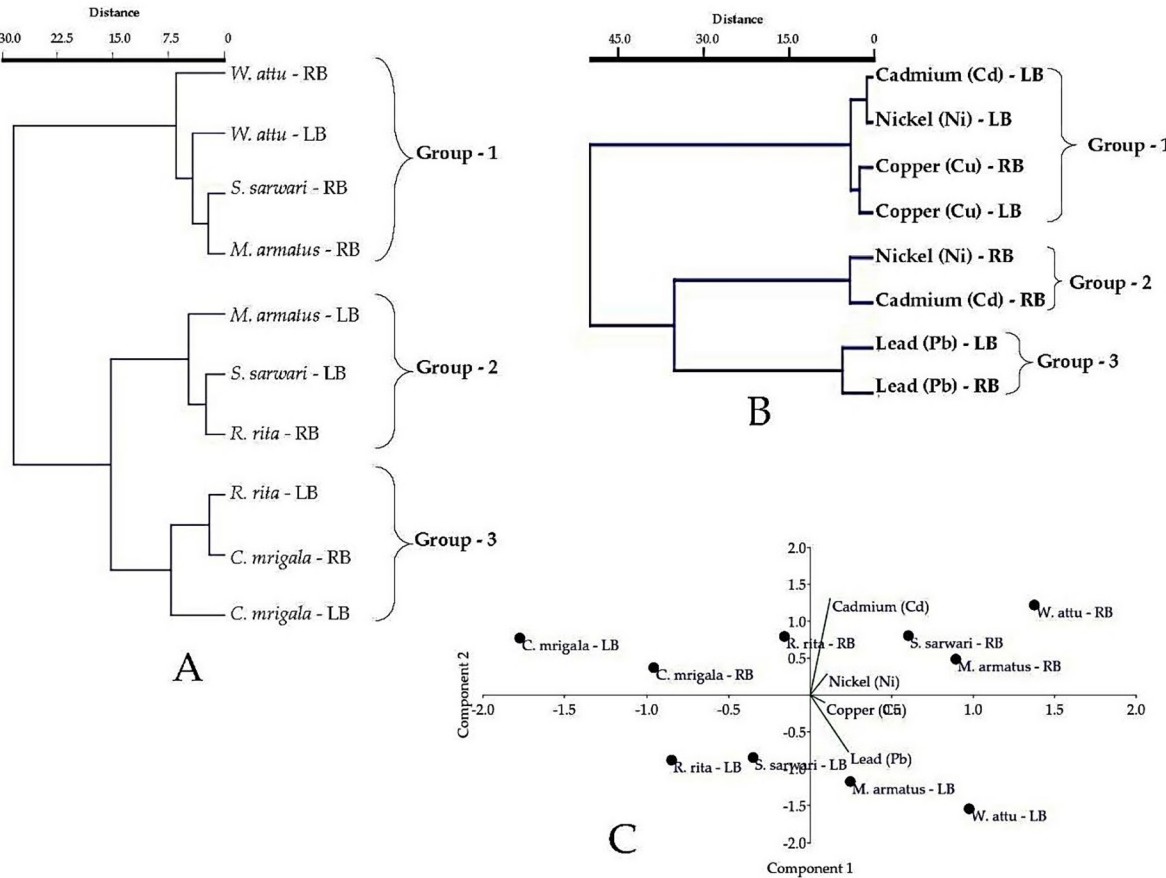

**Fig 5. Multivariate analysis of TTEs (Cd, Cu, Pb, and Ni) in five fish Species (*R. rita*, *S. sarwari*, *W. attu*, *M. armatus*, and *C. mrigala*) at right (RB) and left (LB) of Punjnad headworks.** A: Hierarchal cluster analysis of fish species; B: Hierarchal cluster analysis of TTEs; C: Principal component analysis (PCA) of TTEs and fish species.

## 3.7 Human health risk assessment

Human health risk assessment of TTEs in fish muscles is important to highlight the bioaccumulation and harmful effects of trace elements like Cd, Cu, Pb, and Ni, which can pose serious health risks to consumers. Understanding the levels of TTEs in fish helps in identifying potential hazards, guiding regulatory limits, and ensuring food safety, thereby protecting public health from long-term exposure to toxic substances. The current study evaluated the potential hazard of TTEs present in fish muscles to the mature regular consumers (MRC), juvenile regular consumers (JRC), mature seasonal consumers (MSC), and juvenile seasonal consumers (JSC). Overall, the results showed that the potential hazard of TTEs was in the order of JRC > MRC > JSC > MSC in all fish species and studied TTEs. The maximum exposure hazard was found to the JRC in *W. attu* for Pb accumulation (0.071), while the minimum exposure hazard was recorded to the MSC in *C. mrigala* for Ni concentration (0.006) in fish muscles.

The trend of potential exposure hazard of TTEs was in the order of Pb > Ni > Cd > Cu for *R. rita*, *S. sarwari*, and *W. attu*. In *M. armatus* and *C. mrigala*, the trend was in the order of Pb > Cd > Cu > Ni. Across species, the exposure hazard of Cd and Ni was in the order of *M. armatus* > *W. attu* > *S. sarwari* > *R. rita* > *C. mrigala*, the potential exposure hazard of Pb

was in the order *W. attu > M. armatus > S. sarwari > R. rita > C. mrigala*, and for Ni the trend was in the order of *W. attu > S. sarwari > M. armatus > R. rita > C. mrigala*.

The analysis of THQ showed that for all fish species, Cd had the highest THQ values across all consumer groups, and it was in the order of *M. armatus > W. attu > S. sarwari > R. rita > C. mrigala*. It possessed more health risks for regular consumers (both adults and juveniles), indicating a significant health risk due to Cd exposure. The THQ values for copper are relatively low across all species and consumer groups, with the order of *M. armatus > W. attu > S. sarwari > R. rita > C. mrigala*, suggesting a lower health risk associated with copper exposure. Lead (Pb) also poses a considerable health risk, particularly to juvenile regular consumers, although its THQ values are generally lower than those for Cd. Its exposure across species was in the order of *W. attu > M. armatus > S. sarwari > R. rita > C. mrigala*. The THQ values for Ni are the lowest, with a trend of *W. attu > S. sarwari > M. armatus > R. rita > C. mrigala*. Overall, the highest THQ was observed in Cd (42.826) in the muscles of *M. armatus*, with the potential hazard to the JRC. The lowest THQ effects were observed in *R. rita* muscles affecting MSC in Cu (0.190). The total hazard quotient (TTHQ) values indicate that regular consumers, especially juveniles, are at a higher risk compared to seasonal consumers. Among the fish species listed, *W. attu* and *M. armatus* show the highest TTHQ values (59.56 and 55.33, respectively), signaling that these species are the most hazardous to consume regularly, followed by *S. sarwari*, *R. rita*, and *C. mrigala* with TTHQ values of 43.3, 40.41, and 34.54 respectively. Metal Pollution Index of TTEs ($MPI_{TTE}$) highlighted the overall contamination level found in *M armatus* and *W. attu* (11.88 and 11.71, respectively, showing significantly more contamination of TTEs, followed by *S. sarwari*, *R. rita*, and *C. mrigala* with the MPI values of 9.56, 8.81, and 7.1 respectively (**Table 8**). The assessment indicates that consuming these fish species, particularly regularly, poses a health risk due to the presence of toxic trace elements, especially Cd and Pb. Regular consumers, particularly juveniles, are more at risk than seasonal consumers.

## 4. Discussion

The current study explored the accumulation of selected TTEs (Cd, Cu, Pb, and Ni) in three organs (liver, gills, and muscles) of five fish species (*R. rita*, *S. sarwari*, *W. attu*, *M. armatus*, and *C. mrigala*, during winter, spring, and summer collected from LB and RB of Punjnad headworks. Furthermore, the health risk assessment of these TTEs to human health was also determined to explore their toxic effects on mature and juvenile consumers. A significantly lower accumulation of Cu was indicated in this study. In accordance with our findings, a study was conducted to measure Cu and Zn concentrations in fish samples from three rivers in the Malakand Division, Pakistan. The muscles of *Mastacembelus armatus* from Chakdara on the river Swat exhibited the highest Cu concentration. The study indicated that Zn concentrations were generally higher than Cu in fish organs [52]. Our results showed that *M. armatus* has a higher affinity to accumulate TTEs. Similar to our findings, a study was conducted to determine the TTEs in three fish species, *M. armatus*, *Channa punctatus, and Glossogobius giuris*, from the Turag River in Bangladesh. The study measured concentrations of Pb, Cd, chromium (Cr), Cu, and iron (Fe) in the fish and found higher concentrations of TTEs (Cd and Pb) in *M. armatus* [53]. Lower concentrations of Cu and Ni in our study are coherent with a study that investigated the bioaccumulation of heavy metals viz., Fe, Ni, manganese (Mn), zinc (Zn), and Cu in various tissues of *M. armatus*, including the gills, liver, kidneys, muscles, and integument, from a rivulet in Kasimpur and concluded lower levels of Cu and Ni accumulation in various organs of the fish [54]. Our study showed a high level of Pb accumulation in fish organs, which was also reported in a study by Naz, Chatha [55], showing a high level of Pb accumulation in various organs of fish collected from the Punjnad headworks. However, in

**Table 8. Human health risk assessment of Toxic Trace Elements (TTEs) present in the muscles of studied fish species.** The human health risk assessment is in the order of *R. rita, S. sarwari, W. attu, M. armatus,* and *C. mrigala*, respectively.

| TTEs | $EXP^r_M$ | $EXP^r_J$ | $EXP^s_M$ | $EXP^s_J$ | $THQ^r_M$ | $THQ^r_J$ | $THQ^s_M$ | $THQ^s_J$ | $TTHQ^r_M$ | $TTHQ^r_J$ | $TTHQ^s_M$ | $TTHQ^s_J$ | $MPI_{TTE}$ |
|---|---|---|---|---|---|---|---|---|---|---|---|---|---|
| Cadmium (Cd) | 0.021 | 0.025 | 0.009 | 0.010 | 21.17 | 25.194 | 8.892 | 10.374 | 33.96 | 40.41 | 14.26 | 16.64 | 8.81 |
| Copper (Cu) | 0.018 | 0.022 | 0.008 | 0.009 | 0.452 | 0.538 | 0.190 | 0.222 | | | | | |
| Lead (Pb) | 0.045 | 0.053 | 0.019 | 0.022 | 11.162 | 13.283 | 4.688 | 5.470 | | | | | |
| Nickel (Ni) | 0.023 | 0.028 | 0.010 | 0.012 | 1.175 | 1.398 | 0.493 | 0.576 | | | | | |
| TTEs | $EXP^r_M$ | $EXP^r_J$ | $EXP^s_M$ | $EXP^s_J$ | $THQ^r_M$ | $THQ^r_J$ | $THQ^s_M$ | $THQ^s_J$ | $TTHQ^r_M$ | $TTHQ^r_J$ | $TTHQ^s_M$ | $TTHQ^s_J$ | MPI |
| Cadmium (Cd) | 0.023 | 0.027 | 0.009 | 0.011 | 22.596 | 26.891 | 9.491 | 11.073 | 36.38 | 43.3 | 15.28 | 17.83 | 9.56 |
| Copper (Cu) | 0.019 | 0.022 | 0.008 | 0.009 | 0.472 | 0.561 | 0.198 | 0.231 | | | | | |
| Lead (Pb) | 0.048 | 0.057 | 0.020 | 0.023 | 11.948 | 14.219 | 5.019 | 5.855 | | | | | |
| Nickel (Ni) | 0.027 | 0.033 | 0.011 | 0.013 | 1.368 | 1.628 | 0.575 | 0.670 | | | | | |
| TTEs | $EXP^r_M$ | $EXP^r_J$ | $EXP^s_M$ | $EXP^s_J$ | $THQ^r_M$ | $THQ^r_J$ | $THQ^s_M$ | $THQ^s_J$ | $TTHQ^r_M$ | $TTHQ^r_J$ | $TTHQ^s_M$ | $TTHQ^s_J$ | MPI |
| Cadmium (Cd) | 0.029 | 0.035 | 0.012 | 0.014 | 29.399 | 34.986 | 12.348 | 14.406 | 46.5 | 55.33 | 19.53 | 22.78 | 11.71 |
| Copper (Cu) | 0.021 | 0.025 | 0.009 | 0.010 | 0.515 | 0.613 | 0.216 | 0.253 | | | | | |
| Lead (Pb) | 0.059 | 0.071 | 0.025 | 0.029 | 14.839 | 17.659 | 6.233 | 7.272 | | | | | |
| Nickel (Ni) | 0.035 | 0.041 | 0.015 | 0.017 | 1.743 | 2.075 | 0.732 | 0.854 | | | | | |
| TTEs | $EXP^r_M$ | $EXP^r_J$ | $EXP^s_M$ | $EXP^s_J$ | $THQ^r_M$ | $THQ^r_J$ | $THQ^s_M$ | $THQ^s_J$ | $TTHQ^r_M$ | $TTHQ^r_J$ | $TTHQ^s_M$ | $TTHQ^s_J$ | MPI |
| Cadmium (Cd) | 0.036 | 0.043 | 0.015 | 0.018 | 35.986 | 42.826 | 15.115 | 17.634 | 50.04 | 59.56 | 21.02 | 24.52 | 11.88 |
| Copper (Cu) | 0.031 | 0.037 | 0.013 | 0.015 | 0.784 | 0.933 | 0.329 | 0.384 | | | | | |
| Lead (Pb) | 0.048 | 0.057 | 0.020 | 0.024 | 12.055 | 14.347 | 5.064 | 5.907 | | | | | |
| Nickel (Ni) | 0.024 | 0.029 | 0.010 | 0.012 | 1.218 | 1.450 | 0.512 | 0.597 | | | | | |
| TTEs | $EXP^r_M$ | $EXP^r_J$ | $EXP^s_M$ | $EXP^s_J$ | $THQ^r_M$ | $THQ^r_J$ | $THQ^s_M$ | $THQ^s_J$ | $TTHQ^r_M$ | $TTHQ^r_J$ | $TTHQ^s_M$ | $TTHQ^s_J$ | MPI |
| Cadmium (Cd) | 0.020 | 0.023 | 0.008 | 0.010 | 19.594 | 23.318 | 8.230 | 9.602 | 29.03 | 34.54 | 12.19 | 14.22 | 7.1 |
| Copper (Cu) | 0.019 | 0.023 | 0.008 | 0.009 | 0.482 | 0.573 | 0.202 | 0.236 | | | | | |
| Lead (Pb) | 0.033 | 0.039 | 0.014 | 0.016 | 8.275 | 9.847 | 3.476 | 4.055 | | | | | |
| Nickel (Ni) | 0.014 | 0.016 | 0.006 | 0.007 | 0.676 | 0.805 | 0.284 | 0.331 | | | | | |

†$EXP^r_M$ = Exposure rate in adult regular consumers, $EXP^r_J$ = Exposure rate in juvenile regular consumers, $EXP^s_M$ = Exposure rate in adult seasonal consumers, $EXP^s_J$ = Exposure rate in juvenile seasonal consumers, $THQ^r_M$ = Target hazard quotient in adult regular consumers, $THQ^r_J$ = Target hazard quotient in juvenile regular consumers, $THQ^s_M$ = Target hazard quotient in adult seasonal consumers, $THQ^s_J$ = Target hazard quotient in juvenile seasonal consumers, $TTHQ^r_M$ = Total Target hazard quotient in adult regular consumers, $TTHQ^r_J$ = Total Target hazard quotient in juvenile regular consumers, $TTHQ^s_M$ = Total Target hazard quotient in adult seasonal consumers, $TTHQ^s_J$ = Total Target hazard quotient in juvenile seasonal consumers MPI = Metal pollution index

contrast to our findings, a recent investigation focused on the bioaccumulation of potentially toxic elements (PTEs) including Cd, cobalt (Co), Cr, Cu, lithium (Li), Ni, Pb, selenium (Se), Zn, and Mn in 12 economically important fish species from the lower Ganges in India found that the average concentrations of these potentially toxic elements (PTEs) were ranked as: Zn > Cu > Mn > Ni > Se > Cr > Pb > Co ~ Li > Cd [49]. This could be explained by the nature of this freshwater as our study site has more stagnant water as compared to this study, which could lead to a different pattern of accumulation for TTEs.

In a relatable study, the bioaccumulation of heavy metals arsenic (As), mercury (Hg), and Cd was investigated in five fish species: *Ctenopharyngodon idella, Oreochromis niloticus, Eutropiichthys vacha, R. rita,* and *S. sarwari* from Head Punjnad, Pakistan. The findings indicated that the metal concentrations were ranked as follows: Cd > As > Hg. Specifically, *R. rita, O. niloticus,* and *C. idella* demonstrated elevated levels of metal accumulation, suggesting that species-specific bioaccumulation patterns may be influenced by their feeding behaviors [56]. Another study demonstrated a higher accumulation of Pb as compared to the Cu and Cd concentrations in various organs of catfish [57]. High levels of Cd and Pb accumulation in fish

organs have also been reported in another study, which concluded that in various fish species, Cd and Pb had the highest accumulation in contrast to other TTEs [58]. Our study reported a higher accumulation of Pb and Cd in comparison to Ni and Cu. Similar findings have been reported in a study that explored the concentration of TTEs in various organs of three edible fish species, *Catla catla*, *W*, *attu*, and *Tilapia nilotica*, and found a higher accumulation of Pb and Cd than Ni and Cu [59]. Ali and Khan [60] has studied the bioaccumulation of Cr, Ni, Cd, and Pb in the carnivorous fish *M. armatus* across various sites in three rivers within the Malakand Division, Pakistan. The study revealed that Pb concentrations in the muscles of *M. armatus* were found to be the highest, followed by Cd, which is also reported in current findings. Furthermore, the liver was found to accumulate.

Fish are particularly prone to heavy metal accumulation due to their efficient absorption of metals through the gills, which are involved in both respiration and excretion. The direct exposure of the gills to water facilitates the uptake of environmental metals, highlighting the importance of monitoring heavy metal levels to assess the health risks associated with fish consumption [21]. A study on heavy metal accumulation in *Sparus aurata* showed that fish liver and gills tend to accumulate more heavy metals as compared to muscles [61]. Our study also indicated a higher accumulation of TTEs in fish liver and gills. The accumulation of heavy metals in fish, particularly in critical organs such as the liver, gills, and muscles, is of significant concern due to the potential health risks posed by consuming contaminated fish. These organs tend to retain varying amounts of heavy metals, underscoring the importance of continuous monitoring of metal levels in fish. Such ongoing surveillance is crucial for raising awareness about the potential dangers associated with consuming fish contaminated with heavy metals [62].

Our study reported an exceeded amount of Cd and Pb for their safe human consumption according to the safety limits recommended by the FAO/WHO, indicating a significant risk of metal exposure, which is in accordance with another study that found that the liver had the highest concentration and TTEs accumulation in muscles exceeded the safety limits recommended by the FAO/WHO, indicating a significant risk of metal [54]. Similarly, Pandey, Pandey [63] also showed exceeded concentrations of Cr, Cd, and Pb in the muscles of *R. rita* in comparison to the limits set by the Food and Agriculture Organization (FAO) and the U.S. Environmental Protection Agency (USEPA). A study on TTE pollution in Red Sea Fish from Nuweiba City, Aqaba Gulf, Egypt, showed an exceeded concentration of Pb estimated by THQ [64]. Similarly, some studies showed that Cd contamination in *Clarias gariepinus and Oreochromis niloticus* was a potential threat for being carcinogenic [65]. Likewise, Abbas, El-Sharkawy [66] also showed cancer risks associated with the consumption of cadmium contaminated fish. In a recent study by Anwarul Hasan, Satter [67], heavy metals including Cu, Zn, Pb, Cd, Ni, and As were analyzed in three common fish species viz., *Systomus sarana*, *Pethia ticto, and M. armatus* from the Shitalakshya river using atomic absorption spectroscopy (AAS). The study found that concentrations of Cu, Zn, Pb, Cd, Ni, and as in these fish exceeded the international safety standards set by the FAO/WHO, the U.S. Food and Drug Administration (USFDA), the Ministry of Food and Livestock (MOFL), and the European Commission (EC). Although the targeted hazard quotient (THQ) values for these metals were within the limits deemed safe for individual exposure, the cumulative hazard index (HI) for all three fish species surpassed acceptable levels, indicating a potential health risk for consumers.

According to Naz, Chatha [55] *C. mrigala* from Punjnad, headworks was found to be unsafe for human consumption due to elevated levels of the total target hazard quotient (TTHQs), which is in coherence with current findings. In contrast to our findings, a study concluded that the metal pollution index (MPI), targeted hazard quotient (THQ), and total target hazard quotient (TTHQs) of TTEs were below 1, indicating minimal health risks associated with fish

consumption from this area total target hazard quotient (TTHQs) [49]. Similarly, in a related study, the contamination levels of major rivers, including the Ravi, Chenab, Kabul, and Indus, were assessed, with a particular focus on their impact on fish and human health. The River Ravi, heavily polluted by industrial and sewage wastewater, was identified as the most contaminated, posing significant risks to aquatic life and human health. In contrast, the Indus River, with its larger water volume and fewer industrial sources, demonstrated better ecosystem health. Fish from the Indus, Chenab, and Jhelum Rivers were deemed safe for human consumption. The study underscored the critical need for effective wastewater treatment to mitigate the harmful effects of heavy metals on both aquatic ecosystems and public health [68]. This difference in the acceptability of fish for human consumption could be due to geographical location and increased human activity in the current study site as compared to the other reported sites. It is important to consider this drastic change in the accumulation of TTEs, and the issue must be dealt with urgency to reduce the accumulation of TTEs in water bodies of areas reported in the current study.

## 5. Conclusions

The findings of this study determine the critical impact of industrial and agricultural runoff on the aquatic ecosystem at the Punjnad Headworks, where toxic trace elements (TTEs) are accumulating in significant quantities. Human health risk assessments indicated that both Cd and Pb posed the most considerable exposure hazards. The study concluded that the right bank of the Punjnad headworks is more heavily contaminated than the left, and fish consumption from both banks is unsafe due to the elevated levels of toxic trace elements. The study highlights an urgent need for remedial actions by policymakers, water quality management departments, and environmental agencies to mitigate TTE pollution and protect both aquatic life and human health. Further studies are suggested to explore the impact of other trace elements on the aquatic ecosystem including water, sediments, and living organisms.

## Supporting information

**S1 Table. Average weight, fork and total lengths of fish sampled from the study sites at LB and RB of Punjnad headworks.**
(DOCX)

## Acknowledgments

We would like to express our gratitude to the Directorate of Fish Hatchery, Bahawalpur, for their assistance in netting and sample collection. Our thanks also go to Mian Nawaz Sharif University of Agriculture for analyzing fish samples. Additionally, we extend our appreciation to the Researchers Supporting Project number (RSPD2025R965), King Saud University, Riyadh, Saudi Arabia, for funding.

## Author Contributions

**Conceptualization:** Saima Naz, Ahmad Manan Mustafa Chatha.

**Data curation:** Qudrat Ullah, Dalia Fouad, Maria Lateef, Ahmad Manan Mustafa Chatha.

**Formal analysis:** Qudrat Ullah.

**Investigation:** Maria Lateef, Muhammad Waqar Hassan.

**Methodology:** Saima Naz, Abdul Qadeer, Muhammad Waqar Hassan.

**Software:** Qudrat Ullah, Ahmad Manan Mustafa Chatha.

**Supervision:** Saima Naz.

**Writing – original draft:** Qudrat Ullah, Dalia Fouad, Abdul Qadeer, Maria Lateef.

**Writing – review & editing:** Saima Naz, Muhammad Waqar Hassan, Ahmad Manan Mustafa Chatha.

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
