## [Decision Letter · Decision Letter 0]

24 Sep 2024

PONE-D-24-39090Trace Elements in Fish: Assessment of bioaccumulation and associated health risks.PLOS ONE

Dear Dr. Ullah,

Thank you for submitting your manuscript to PLOS ONE. After careful consideration, we feel that it has merit but does not fully meet PLOS ONE’s publication criteria as it currently stands. Therefore, we invite you to submit a revised version of the manuscript that addresses the points raised during the review process.

We look forward to receiving your revised manuscript.

Kind regards,

Ulaganathan Arisekar

Academic Editor

PLOS ONE

**Journal Requirements:**

2. To comply with PLOS ONE submissions requirements, in your Methods section, please provide additional information regarding the experiments involving animals and ensure you have included details on (a) methods of sacrifice, (b) methods of anesthesia and/or analgesia, and (c) efforts to alleviate suffering.

4. In your Methods section, please provide additional information regarding the permits you obtained for the work. Please ensure you have included the full name of the authority that approved the field site access and, if no permits were required, a brief statement explaining why.

7. We note that Figure 1 in your submission contain map images which may be copyrighted. All PLOS content is published under the Creative Commons Attribution License (CC BY 4.0), which means that the manuscript, images, and Supporting Information files will be freely available online, and any third party is permitted to access, download, copy, distribute, and use these materials in any way, even commercially, with proper attribution. For these reasons, we cannot publish previously copyrighted maps or satellite images created using proprietary data, such as Google software (Google Maps, Street View, and Earth). For more information, see our copyright guidelines: http://journals.plos.org/plosone/s/licenses-and-copyright.

We require you to either present written permission from the copyright holder to publish these figures specifically under the CC BY 4.0 license, or remove the figures from your submission:

8. We notice that your supplementary tables are included in the manuscript file. Please remove them and upload them with the file type 'Supporting Information'. Please ensure that each Supporting Information file has a legend listed in the manuscript after the references list.

Reviewers' comments:

Reviewer's Responses to Questions

**Comments to the Author**

1. Is the manuscript technically sound, and do the data support the conclusions?

Reviewer #1: No

Reviewer #2: Yes

Reviewer #3: Yes

Reviewer #4: Yes

2. Has the statistical analysis been performed appropriately and rigorously? 

Reviewer #1: No

Reviewer #2: Yes

Reviewer #3: Yes

Reviewer #4: Yes

3. Have the authors made all data underlying the findings in their manuscript fully available?

Reviewer #1: Yes

Reviewer #2: No

Reviewer #3: Yes

Reviewer #4: Yes

4. Is the manuscript presented in an intelligible fashion and written in standard English?

Reviewer #1: No

Reviewer #2: Yes

Reviewer #3: No

Reviewer #4: Yes

5. Review Comments to the Author

**Reviewer #1:** Manuscript #: PONE-D-24-39090

Title: Trace Elements in Fish: Assessment of bioaccumulation and associated health risks.

Comment 1: The authors have not provided a detailed explanation of the sampling methodology used to collect fish samples from the Left and Right banks of Punjnad headworks. Specifically, what was the spatial distribution of the sampling sites, and how were the fish species selected for this study?

Comment 2: Can the authors provide a comprehensive analysis of the physicochemical parameters of the water samples collected from the Left and Right banks of Punjnad headworks during the three seasons (winter, spring, and summer)? How do these parameters influence the bioaccumulation of TTEs in fish?

Comment 3: The study has only investigated the accumulation of four TTEs (Cd, Cu, Pb, and Ni) in three organs (liver, gills, and muscles) of five fish species. However, there are other essential trace elements, such as zinc (Zn), iron (Fe), and manganese (Mn), that are also important for metabolic processes in organisms. Would the inclusion of these elements have altered the results of the study?

Comment 4: How do the authors explain the significant accumulation of Pb in the liver of fish, considering that Pb is a non-essential element with no known biological function? Is there a possibility of Pb replacing essential elements like calcium (Ca) or zinc (Zn) in the liver, leading to toxicity?

Comment 5: The human health risk assessment was based on the estimated exposure hazards, hazardous index (THQ and TTHQ), and metal pollution index (MPI). However, the authors have not provided a detailed explanation of the assumptions and limitations of these indices. How do these indices account for the variability in fish consumption patterns among different age groups and populations?

Comment 6: Can the authors provide a probabilistic analysis of the human health risks associated with consuming fish from the Left and Right banks of Punjnad headworks? How do the results of this study inform policy decisions regarding fish consumption advisories and water quality management in the region?

Comment 7: The study has not explored the potential interactions between TTEs and other environmental pollutants, such as pesticides, industrial effluents, and agricultural runoff, that may influence the bioaccumulation of TTEs in fish. How do these interactions affect the accuracy of the human health risk assessment?

Comment 8: Are there any ongoing or planned monitoring programs to track changes in TTE concentrations in fish and water samples from the Left and Right banks of Punjnad headworks? How can the results of this study inform the development of early warning systems for TTE pollution in the region?

Comment 9: The authors have not discussed the potential implications of their study for aquaculture development and fisheries management in the region. How can the results of this study inform the development of sustainable aquaculture practices and fishery management strategies that minimize the risks associated with TTE pollution?

Comment 10: Can the authors provide a cost-benefit analysis of implementing measures to reduce TTE pollution in the Punjnad headworks, such as wastewater treatment infrastructure, pollution control technologies, and environmental restoration programs? How do the benefits of these measures outweigh the costs in terms of human health, environmental sustainability, and economic development?

Comment 11: “These toxic trace elements (TTEs) enter into water through waste from industries like tanneries, textiles, metal finishing, mining, dyeing, ceramics, and pharmaceuticals” Add the following reference:

2018. High-efficiency extraction of bromocresol purple dye and heavy metals as chromium from industrial effluent by adsorption onto a modified surface of zeolite: kinetics and equilibrium study. Journal of Environmental Management 225, 120–132.

Comment 12: The study area selection seems to be biased towards the Punjnad headworks, which may not be representative of the entire Chenab river. How did the authors ensure that the selected sites are truly representative of the entire region, and what were the criteria used to exclude other potential sites?

Comment 13: What was the rationale behind selecting only five fish species for this study, and how do the authors justify the exclusion of other species that may be more sensitive to trace element bioaccumulation?

Comment 14: The use of a gauze net with a mesh size of 60 mm may have resulted in the exclusion of smaller fish species or juveniles, which could have affected the results. Can the authors provide data on the size distribution of the fish caught and justify the mesh size selection?

Comment 15: How did the authors account for potential sources of contamination during sample collection, storage, and transportation, and what quality control measures were implemented to ensure the integrity of the samples?

Comment 16: The acid digestion method used may have introduced biases in the analysis, particularly for certain elements like chromium. Have the authors considered alternative methods, such as microwave-assisted digestion or enzymatic digestion, and how do they justify their choice of method?

Comment 17: “The accumulation of TTEs in fish mainly depends upon the concentration of these metals in the aquatic environment and the duration of exposure” and “Both fish and humans are primarily exposed to lead through ingestion and inhalation” Add the following reference:

2022. Evaluation of concentrations of trace metal(loid)s in indigenous crab species and human health risk implications. Journal of Emerging Contaminants 8, 371-380.

Comment 18: The detection limits for the elements analyzed seem relatively high compared to other studies. Can the authors provide a detailed justification for the instrument settings used and the detection limits chosen?

Comment 19: The study only analyzed the liver, gills, and muscles of the fish, which may not provide a comprehensive picture of trace element bioaccumulation. How do the authors justify the exclusion of other tissues, such as the kidneys or bones, which may also be affected by trace element exposure?

Comment 20: Have the authors considered the potential effects of seasonal variations in water temperature, pH, and other environmental factors on trace element bioaccumulation, and how did they account for these factors in their analysis?

Comment 21: The manuscript does not provide adequate information on the statistical analysis used to evaluate the data. Can the authors provide a detailed description of the statistical methods employed, including the tests used to compare means and the corrections made for multiple comparisons?

Comment 22: How do the authors plan to communicate the findings of this study to stakeholders, including policymakers, farmers, and local communities, to ensure that the results are translated into effective mitigation strategies to reduce trace element contamination in the Chenab river?

Comment 23: “Some trace elements, like zinc (Zn), copper (Cu), iron (Fe), and manganese (Mn), are essential for metabolic processes in organisms but exist in a narrow range between being beneficial and toxic” Add the following reference:

2021. Comparative study of the biochemical response behavior of some highly toxic minerals on selenosis in rats. Revista de Chimie 72(2), 9-18.

Comment 24: Critical Review of the Health Risk Assessment – Although, the authors have made an attempt to assess the health risks associated with the consumption of fish contaminated with toxic trace elements (TTEs), there are several limitations and concerns that need to be addressed. Firstly, the ingestion exposure estimation equations (Equations 1-4) assume a constant ingestion rate of TTEs, which may not be realistic considering the variability in fish consumption patterns among different populations. How do the authors account for the potential variations in ingestion rates among individuals, particularly in regions where fish is a staple food? Secondly, the use of a single reference dose (RfD) value for each TTE may not be sufficient to capture the complexity of the toxicological effects of these elements. For instance, the RfD value of 0.001 mg/kg/day for Cd is based on the assumption of a linear dose-response relationship, which may not hold true at higher exposure levels. Can the authors provide a more detailed justification for the selection of these RfD values, and how they account for potential non-linear effects? Thirdly, the calculation of the Total Target Hazardous Quotient (TTHQ) assumes that the effects of individual TTEs are additive, which may not be the case in reality. How do the authors account for potential synergistic or antagonistic interactions between different TTEs, which could influence the overall health risk? Fourthly, the Metal Pollution Index (MPI) calculation (Equation 13) is based on a simplistic multiplication of the concentrations of individual TTEs, which may not accurately reflect the cumulative effects of these elements. Can the authors provide a more detailed explanation of the theoretical basis for this index, and how it accounts for the varying toxicological potencies of different TTEs?

Comment 25: How do the authors plan to validate the accuracy of their health risk assessment model using real-world data from fish consumers?

Comment 26: What is the sensitivity of the TTHQ and MPI calculations to changes in the input parameters, such as the RfD values or ingestion rates?

Comment 27: Can the authors provide a more detailed analysis of the bioaccumulation patterns of individual TTEs in different fish species, and how these patterns influence the health risk assessment?

Comment 28: How do the authors plan to extend their study to include other potential sources of TTE exposure, such as drinking water or soil contamination?

Comment 29: (1) Study the development of more sophisticated health risk assessment models that account for the variability in ingestion rates, exposure frequencies, and toxicological effects of individual TTEs. (2) Investigate the potential interactions between different TTEs, and how these interactions influence the overall health risk. (3) Perform the validation of the MPI calculation using real-world data from fish consumers, and exploration of alternative indices that better capture the cumulative effects of TTEs. (4) Extend the study to include other potential sources of TTE exposure, and development of a comprehensive framework for assessing the health risks associated with TTE contamination in the environment.

Comment 30: The ingestion exposure estimation (Equations 1-4) relies on several assumptions, which may lead to uncertainty in the results. For instance, the ingestion rate of mature (IRM) and juvenile (IRJ) persons is assumed to be constant, which may not reflect the variability in consumption patterns among different populations. Moreover, the exposure frequency (EFs and EFr) and duration (EDM and EDJ) are based on averages, which may not account for individual differences in consumption habits.

Comment 31: “Other trace elements such as cadmium (Cd), mercury (Hg), chromium (Cr), and lead (Pb), can be extremely toxic even at low concentrations, especially under certain conditions, making regular monitoring of sensitive aquatic environments necessary”, “THQ represents the Hazard Quotient, calculated based on ingestion at the corresponding exposure level; RfD represents the reference dose for the potential hazardous health effects caused by contaminants through ingestion of TTEs”, and “The Total target hazardous quotient (TTHQ) is the sum of THQs for individual TTE and represent the cumulative exposure effects of all TTEs to human health” Add the following reference:

2024. Evaluation and risk assessment of heavy metals in King tuber mushroom in the contest of COVID-19 pandemic lockdown in Sokoto state, Nigeria. Kuwait Journal of Science 51(2), 100193.

Comment 32: The target hazardous quotients (THQs) estimation (Equations 5-8) is based on the reference dose (RfD) values, which may not be representative of the entire population. The RfD values used in the study (0.001, 0.04, 0.004, and 0.02 for Cd, Cu, Pb, and Ni, respectively) are based on average values, which may not account for individual variations in susceptibility to TTEs.

Comment 33: The total target hazardous quotient (TTHQ) estimation (Equations 9-12) assumes an additive effect of individual TTEs, which may not account for potential synergistic effects. The interaction between different TTEs can lead to enhanced toxicity, which is not considered in the current study.

Comment 34: The metal pollution index (MPI) calculation (Equation 13) is based on the geometric mean of the concentrations of individual TTEs, which may not accurately reflect the overall pollution level. The MPI does not consider the bioavailability and bioaccessibility of TTEs, which can affect their toxicity.

Comment 35: The statistical analysis using one-way ANOVA and post hoc Duncan multiple range test may not account for the complexity of the data. The use of Pearson's correlation analysis and principal component analysis (PCA) may not fully capture the relationships between fish species and TTEs.

Comment 36: “Contaminated irrigation water can significantly impact crops grown along the riverbanks, posing serious risks to agricultural productivity and food safety” Add the following reference:

2023. Food products quality and nutrition in relation to public. Balancing health and disease. Progress in Nutrition 25(1), e2023024.

Comment 37: (1) Develop individual-based models that account for variability in consumption patterns, exposure frequency, and duration to improve the accuracy of ingestion exposure estimation. (2) Investigate the synergistic effects of TTEs on human health to develop a more comprehensive understanding of their toxicity. (3) Incorporate bioavailability and bioaccessibility assessments into the MPI calculation to better reflect the toxicity of TTEs. (4) Employ advanced statistical techniques, such as machine learning algorithms, to better capture the relationships between fish species and TTEs.

Comment 38: How do the assumptions made in the ingestion exposure estimation affect the accuracy of the results?

Comment 39: Can the THQs estimation be improved by incorporating individual variations in susceptibility to TTEs?

Comment 40: How can the synergistic effects of TTEs be quantified and incorporated into the TTHQ estimation?

Comment 41: What are the limitations of using the MPI as a measure of metal pollution, and how can it be improved?

By addressing these comments, the authors can strengthen the validity and generalizability of their study, and provide more comprehensive insights into the bioaccumulation of trace elements in fish and the associated health risks.

**Reviewer #2:** Comments to the Author

Paper has a good structure, well defined titles and subtitles. The presented results are very well organized and detailed.

Abstract is well written, and useful.

The Introduction section is informative, with adequate cited reference, very well written, hypotheses are well appointed.

Materials and methods section is detailed written and extensive.

The section Results is good structured supported with good organized

Conclusion section is written very clear, concise, with some further activities and warnings.

Minor comments

L 132 – correct the sentence

L135 – explain the abbreviations RB and LB

Correct the map – it is illegible

Table 1 – add the Limit Of Detection

L177 use the metric system (SI)

How many fish were taken for testing?

What was the age of the fish.

Add a short description of the food preferences of the species tested.

Why was TTE not determined in water? Maybe monitoring studies are available?

Supplementatry Table 1 – were the differences significant?

L 331: How can the water flow affect metal concentrations in fish?

RfD for Pb is 1.43x10-4, see: Li Z, Ma Z, van der Kuijp TJ, Yuan Z, Huang L. 2014. A review of soil heavy metal pollution from mines in China: pollution and health risk assessment. Science of The Total Environment 468–469:843–853. https://doi.org/10.1016/j.scitotenv.2013.08.090

Section 2.4.2.: Take into account new guidelines: U.S. E.P.A. United States Environmental Protection Agency, 2023. Regional Screening Level (RSL) summary table (TR=1E-06, HQ=1) November 2023. https://semspub.epa.gov/work/HQ/400431.pdf

**Reviewer #3: **Comments

I have read the manuscript PONE-D-24-39090: “Trace Elements in Fish: Assessment of bioaccumulation and associated health risks.” for possible publication.

The following points may be considered while revising the article:

1. Please revise the short title and make it more attractive.

2. The keywords are not suitable. Avoid using the exact words already being used in the title.

3. The authors should improve the introduction section and add more details about the rationale of this study.

4. The abstract is not well written and is mostly about methods. Please carefully revise the abstract.

5. The introduction is very lengthy, so please revise and shorten it.

6. Please improve the rationale of this study.

7. The materials and methods section is well written.

8. There are lots of English grammatical mistakes, so revise the whole manuscript for such mistakes.

9. The results section lacks clarity and organization. It would be better to restructure the results section to improve readability and make it easier for readers.

10. Please improve the conclusion section.

**Reviewer #4:** Review Comments to the Author

Manuscript Number: PONE-D-24-39090

Dear authors,

Peace be upon you and God's mercy

Hoping you are fine.

Thanks for submitting your manuscript "Trace Elements in Fish: Assessment of Bioaccumulation and Associated Health Risks" to PLOS ONE. I noticed that the current study aimed to investigate the bioaccumulation of four toxic trace elements (Cadmium (Cd), Copper (Cu), Lead (Pb), and Nickel (Ni)) in three organs (liver, gills, and muscles) of five fish species, Rita rita , Sperata sarwari , Wallago attu , Mastacembelus armatus , and Cirrhinus mrigala , collected from both the right and left banks of Punjnad Headworks during the winter, spring, and summer seasons. The study also assesses the potential human health risks by calculating the exposure hazards, Target Hazard Quotient (THQ), Total Target Hazard Quotient (TTHQ), and Metal Pollution Index (MPI).

The results demonstrated that Wallago attu accumulated significantly higher levels of TTEs (p < 0.00) compared to the other species. Among the seasons, summer exhibited the highest accumulation of TTEs. Lead (Pb) showed the highest concentration across all TTEs, particularly in the liver compared to the gills and muscles. Additionally, the right bank displayed significantly higher TTE accumulation (p < 0.00) than the left bank across all fish species. The human health risk assessment indicated that Cd and Pb posed the greatest exposure risks, with THQ values for these metals exceeding safe limits in all species. The MPI values also indicated moderate to high levels of contamination. The study concluded that fish from both banks of the Punjnad Headworks pose significant health risks for human consumption.

Revisions Needed:

After reviewing the manuscript, several areas need improvement:

- Title: The current title is too broad. It would benefit from specifying either the fish species or the geographic location to enhance clarity and focus.

Suggested Titles:

1. "Trace Elements in Fish Species from the Punjnad Headworks: Bioaccumulation and Human Health Risk Assessment"

2. "Bioaccumulation of Toxic Trace Elements in Fish from Industrial and Agricultural Runoff: A Study at Punjnad Headworks"

- Keywords: The keywords require revision to more accurately reflect the scope of the study, including specific references to the toxic trace elements, the fish species, and the geographic context.

- Materials and Methods: This section requires more detailed descriptions, especially concerning the analytical methods used to measure the trace elements. Specific details regarding the sampling procedures, laboratory techniques, and statistical analyses should be included to provide a clearer understanding of the methodology.

- Discussion: The discussion section needs significant enhancement. The current version lacks depth in linking the findings to existing literature and prior research. By incorporating a more comprehensive literature review, the analysis can be more robust and provide stronger conclusions. The provided references can help support and develop this section further.

Recommendations for Improvement:

1. Revise the title to reflect the specific fish species or geographic location.

2. Refine the keywords to better capture the study's main elements (e.g., toxic trace elements, fish bioaccumulation, human health risks, Punjnad Headworks).

3. Expand the materials and methods section by detailing the sampling and analytical procedures.

4. Strengthen the discussion by incorporating more recent and relevant literature, using the following references:

• https://doi.org/10.1007/s12011-023-03880-0

• https://doi.org/10.1007/s12011-024-04246-w

• https://doi.org/10.1016/j.aquaculture.2024.740833

• https://dx.doi.org/10.21608/ejabf.2024.336220

• https://doi.org/10.1007/s12011-023-04007-1

• https://dx.doi.org/10.21608/ejabf.2023.314426

Major revisions are required to bring the manuscript in line with current scientific standards and to improve the quality of the data analysis and presentation.

6. PLOS authors have the option to publish the peer review history of their article (what does this mean?). If published, this will include your full peer review and any attached files.

Reviewer #1: **Yes: **Loai Aljerf

Reviewer #2: No

Reviewer #3: No

Reviewer #4: No

---

## [Author Response · Author response to Decision Letter 0]

18 Oct 2024

Rebuttal Letter

We would like to express our sincere gratitude to the reviewers for their time and effort in reviewing our manuscript. Your insightful and constructive comments have been invaluable in improving the quality of our work. We appreciate your thoughtful suggestions and have carefully considered each one in our revisions. Your expertise has helped us refine the content, and we believe the manuscript is now stronger as a result. Thank you again for your dedication and for providing us with the opportunity to enhance our research. The responses to the individual comments of each reviewer are provided below. 

Reviewer 1 

Comment 1: The authors have not provided a detailed explanation of the sampling methodology used to collect fish samples from the Left and Right banks of Punjnad headworks. Specifically, what was the spatial distribution of the sampling sites, and how were the fish species selected for this study?

Response 1: The suggested information is added to the manuscript at line # 157-158 and 179-178.

Comment 2: Can the authors provide a comprehensive analysis of the physicochemical parameters of the water samples collected from the Left and Right banks of Punjnad headworks during the three seasons (winter, spring, and summer)? How do these parameters influence the bioaccumulation of TTEs in fish?

Response 2: a comprehensive analysis of the physicochemical parameters of the water samples collected from the Left and Right banks of Punjnad headworks during the three seasons has been added to the manuscript.

Comment 3: The study has only investigated the accumulation of four TTEs (Cd, Cu, Pb, and Ni) in three organs (liver, gills, and muscles) of five fish species. However, there are other essential trace elements, such as zinc (Zn), iron (Fe), and manganese (Mn), that are also important for metabolic processes in organisms. Would the inclusion of these elements have altered the results of the study?

Response 3: Due to the limited available funds and resources, only four TTEs (Cd, Cu, Pb, and Ni) were included in the study. It is included in the manuscript at line # 174-175. However, as the selected trace elements are most toxic trace elements in the study area, inclusion of other trace element should not have affected the results. Still, it can only be confirmed by conducting experiments. We will consider this in future studies. 

Comment 4: How do the authors explain the significant accumulation of Pb in the liver of fish, considering that Pb is a non-essential element with no known biological function? Is there a possibility of Pb replacing essential elements like calcium (Ca) or zinc (Zn) in the liver, leading to toxicity?

Response 4: The significantly higher accumulation of lead in fish samples indicates a high level of industrial and agricultural activities in the vicinity of the study site. Which resulted in the accumulation of Pb in the water. Liver being the detoxification center of the body, accumulated more Pb. In our opinion, it does not show the possibility of replacement of any of the essential trace elements in fish body. 

Comment 5: The human health risk assessment was based on the estimated exposure hazards, hazardous index (THQ and TTHQ), and metal pollution index (MPI). However, the authors have not provided a detailed explanation of the assumptions and limitations of these indices. How do these indices account for the variability in fish consumption patterns among different age groups and populations? 

Response 5: The suggested details on assumptions and limitations of indices used for the human health risk assessment is added to the manuscript at line # 235-251.

Comment 6: Can the authors provide a probabilistic analysis of the human health risks associated with consuming fish from the Left and Right banks of Punjnad headworks? How do the results of this study inform policy decisions regarding fish consumption advisories and water quality management in the region?

Response 6: Providing a probabilistic analysis of the human health risks associated with fish consumption is at this stage is not possible. However, the message of the study to the policy makers and government agencies like water quality management has been added to the manuscript at line # 664-665.

Comment 7: The study has not explored the potential interactions between TTEs and other environmental pollutants, such as pesticides, industrial effluents, and agricultural runoff, that may influence the bioaccumulation of TTEs in fish. How do these interactions affect the accuracy of the human health risk assessment?

Response 7: We had very limited resources to conduct the experiment. Therefore, it was not possible to conduct further interactions of study TTEs with other environmental pollutants such as pesticides, industrial effluents, and agricultural runoff, that may influence the bioaccumulation of TTEs in fish. The trace elements explored in this study are the most abundant TTEs and are the results of exposure of various effluences from industrial, agricultural, domestic pollution sources. The interaction between these sources could have improved the understanding of accumulation of TTEs but these interactions would not affect the results significantly. 

Comment 8: Are there any ongoing or planned monitoring programs to track changes in TTE concentrations in fish and water samples from the Left and Right banks of Punjnad headworks? How can the results of this study inform the development of early warning systems for TTE pollution in the region? 

Response 8: No. there is no ongoing or planned monitoring program to track changes in TTEs concentrations in fish and water samples from the Left and Right banks of Punjnad headworks. This study has emphasized the relevant agencies for quick and effective countermeasures to address these issues at the study sites. 

Comment 9: The authors have not discussed the potential implications of their study for aquaculture development and fisheries management in the region. How can the results of this study inform the development of sustainable aquaculture practices and fishery management strategies that minimize the risks associated with TTE pollution?

Response 9: Being an early investigation, the current study requires further exploration of the target sites for other pollutants and their interaction with fish, other aquatic species, and human health to fully address the potential impact and condition of aquatic pollution at the target site. Only then any viable action plan can be developed for the sustainable aquaculture practices and fishery management strategies that minimize the risks associated with TTE pollution at the target sites. 

Comment 10: Can the authors provide a cost-benefit analysis of implementing measures to reduce TTE pollution in the Punjnad headworks, such as wastewater treatment infrastructure, pollution control technologies, and environmental restoration programs? How do the benefits of these measures outweigh the costs in terms of human health, environmental sustainability, and economic development?

Response 10: As explained before, being an early investigation, the current study requires further exploration of the target sites for other pollutants and their interaction with fish, other aquatic species, and human health to provide a cost-benefit analysis of implementing measures to reduce TTE pollution in the Punjnad headworks. 

Comment 11: “These toxic trace elements (TTEs) enter into water through waste from industries like tanneries, textiles, metal finishing, mining, dyeing, ceramics, and pharmaceuticals” Add the following reference:2018. High-efficiency extraction of bromocresol purple dye and heavy metals as chromium from industrial effluent by adsorption onto a modified surface of zeolite: kinetics and equilibrium study. Journal of Environmental Management 225, 120–132.

Response 11: The suggested reference has been added in the manuscript at line # 60.

Comment 12: The study area selection seems to be biased towards the Punjnad headworks, which may not be representative of the entire Chenab River. How did the authors ensure that the selected sites are truly representative of the entire region, and what were the criteria used to exclude other potential sites?

Response 12: The current study assessed the TTE accumulation only at the region of Punjnad headworks, and does not target or represent the entire region of Chenab River. 

Comment 13: What was the rationale behind selecting only five fish species for this study, and how do the authors justify the exclusion of other species that may be more sensitive to trace element bioaccumulation?

Response 13: The fish species were selected on the basis of abundance during netting. Due to limited resources, it was not possible to select all the available species at the target site. The rationale explanation is provided in the manuscript at line # 177-180.

Comment 14: The use of a gauze net with a mesh size of 60 mm may have resulted in the exclusion of smaller fish species or juveniles, which could have affected the results. Can the authors provide data on the size distribution of the fish caught and justify the mesh size selection?

Response 14: The target sites were the property of Government under the department of Fisheries, Punjab, Pakistan. We were only allowed to sample adult fish using a net with no less than 60 mm mesh size. Furthermore, the fishermen do not use juvenile fish for the consumers as it is not allowed. So, choosing the given mesh size was a requirement and had no effect on the results of this study. 

Comment 15: How did the authors account for potential sources of contamination during sample collection, storage, and transportation, and what quality control measures were implemented to ensure the integrity of the samples?

Response 15: The suggested text is added to the manuscript at line # 184-188, and 195-196. 

Comment 16: The acid digestion method used may have introduced biases in the analysis, particularly for certain elements like chromium. Have the authors considered alternative methods, such as microwave-assisted digestion or enzymatic digestion, and how do they justify their choice of method?

Response 16: The laboratory resources at the University do not have the option for alternative methods for digestion. The method used is applied by the expert and with utmost care to avoid any biasness for metal detection. 

Comment 17: “The accumulation of TTEs in fish mainly depends upon the concentration of these metals in the aquatic environment and the duration of exposure” and “Both fish and humans are primarily exposed to lead through ingestion and inhalation” Add the following reference:

Response 16: The suggested reference has been added at line # 100. 

Comment 18: The detection limits for the elements analyzed seem relatively high compared to other studies. Can the authors provide a detailed justification for the instrument settings used and the detection limits chosen? 

Response 18: The detailed settings are provided in the table 1. 

Comment 19: The study only analyzed the liver, gills, and muscles of the fish, which may not provide a comprehensive picture of trace element bioaccumulation. How do the authors justify the exclusion of other tissues, such as the kidneys or bones, which may also be affected by trace element exposure?

Response 19: Due to limited available funds and resources, we analyzed only three organs including fish muscles which is the consumable organ and liver and gills being important for higher accumulation of trace element. 

Comment 20: Have the authors considered the potential effects of seasonal variations in water temperature, pH, and other environmental factors on trace element bioaccumulation, and how did they account for these factors in their analysis? 

Response 20. Seasonal variations in the water parameters have significant impact on bioaccumulation. This is the reason to include the seasonal sampling in the study. Furthermore, the effects of seasonal variations on trace elements in water at the study site has already been investigated by the principal author. “Naz, S., Mansouri, B., Chatha, A.M.M. et al. Water quality and health risk assessment of trace elements in surface water at Punjnad Headworks, Punjab, Pakistan. Environ Sci Pollut Res 29, 61457–61469 (2022). https://doi.org/10.1007/s11356-022-20210-4”. However, due to limitation of resources we cannot include these analyses in the current study. 

Comment 21: The manuscript does not provide adequate information on the statistical analysis used to evaluate the data. Can the authors provide a detailed description of the statistical methods employed, including the tests used to compare means and the corrections made for multiple comparisons?

Response 21. The required information is updated in the manuscript at line # 310-314.

Comment 22: How do the authors plan to communicate the findings of this study to stakeholders, including policymakers, farmers, and local communities, to ensure that the results are translated into effective mitigation strategies to reduce trace element contamination in the Chenab river?

Response 22. As already stated, this is a preliminary work at the target site. After further analysis of other important trace elements and fish species the results will be shared with the policy makers, farmers and local communities to reduce the impact of TTEs and improve the well being of aquatic animals and human consumers. 

Comment 23: “Some trace elements, like zinc (Zn), copper (Cu), iron (Fe), and manganese (Mn), are essential for metabolic processes in organisms but exist in a narrow range between being beneficial and toxic” Add the following reference:2021. Revista de Chimie 72(2), 9-18.

Response 23: The suggested reference has been added at line # 100. 

Comment 24: Critical Review of the Health Risk Assessment – Although, the authors have made an attempt to assess the health risks associated with the consumption of fish contaminated with toxic trace elements (TTEs), there are several limitations and concerns that need to be addressed. Firstly, the ingestion exposure estimation equations (Equations 1-4) assume a constant ingestion rate of TTEs, which may not be realistic considering the variability in fish consumption patterns among different populations. How do the authors account for the potential variations in ingestion rates among individuals, particularly in regions where fish is a staple food? Secondly, the use of a single reference dose (RfD) value for each TTE may not be sufficient to capture the complexity of the toxicological effects of these elements. For instance, the RfD value of 0.001 mg/kg/day for Cd is based on the assumption of a linear dose-response relationship, which may not hold true at higher exposure levels. Can the authors provide a more detailed justification for the selection of these RfD values, and how they account for potential non-linear effects? Thirdly, the calculation of the Total Target Hazardous Quotient (TTHQ) assumes that the effects of individual TTEs are additive, which may not be the case in reality. How do the authors account for potential synergistic or antagonistic interactions between different TTEs, which could influence the overall health risk? Fourthly, the Metal Pollution Index (MPI) calculation (Equation 13) is based on a simplistic multiplication of the concentrations of individual TTEs, which may not accurately reflect the cumulative effects of these elements. Can the authors provide a more detailed explanation of the theoretical basis for this index, and how it accounts for the varying toxicological potencies of different TTEs?

Response 24: This is an excellent point raised by the respected reviewer. As the health risk assessment is done on the basis of various estimation indices certain limitations and assumption are associated with the resultant analysis. These assumptions and limitations have been added into the manuscript with suitable references at line # 240-256. 

Comment 25: How do the authors plan to validate the accuracy of their health risk assessment model using real-world data from fish consumers?

Response 25: After further analysis of other fish species and TTEs at the target site, we aim to conduct epidemiological studies, Biomo

---

## [Decision Letter · Decision Letter 1]

27 Dec 2024

Trace Elements in Fish Species from the Punjnad Headworks: Bioaccumulation and Human Health Risk Assessment

PONE-D-24-39090R1

Dear Dr. Qudrat Ullah

We’re pleased to inform you that your manuscript has been judged scientifically suitable for publication and will be formally accepted for publication once it meets all outstanding technical requirements.

Kind regards,

Ulaganathan Arisekar

Academic Editor

PLOS ONE

Additional Editor Comments (optional):

All the comments are carried out meticulously

Reviewers' comments:

Reviewer's Responses to Questions

**Comments to the Author**

1. If the authors have adequately addressed your comments raised in a previous round of review and you feel that this manuscript is now acceptable for publication, you may indicate that here to bypass the “Comments to the Author” section, enter your conflict of interest statement in the “Confidential to Editor” section, and submit your "Accept" recommendation.

Reviewer #2: All comments have been addressed

Reviewer #4: All comments have been addressed

2. Is the manuscript technically sound, and do the data support the conclusions?

Reviewer #2: Partly

Reviewer #4: (No Response)

3. Has the statistical analysis been performed appropriately and rigorously? 

Reviewer #2: Yes

Reviewer #4: Yes

4. Have the authors made all data underlying the findings in their manuscript fully available?

Reviewer #2: Yes

Reviewer #4: Yes

5. Is the manuscript presented in an intelligible fashion and written in standard English?

Reviewer #2: Yes

Reviewer #4: (No Response)

6. Review Comments to the Author

Reviewer #2: I read your revised version of the manuscript and your answers to my points. many of the mistakes are eliminated.

Authors should correct typos.

Reviewer #4: Reviewer Report

Manuscript Title: Trace Elements in Fish Species from the Punjnad Headworks: Bioaccumulation and Human Health Risk Assessment

Manuscript Number: PONE-D-24-39090R1

The authors have addressed all the comments and suggestions provided in the initial review thoroughly and effectively. Significant improvements were made to the abstract, introduction, methodology, results, and discussion sections, which have enhanced the clarity and scientific quality of the manuscript.

The research objectives are now clearly defined, and the methodology is well-described with appropriate analytical tools. The results are presented systematically and are supported by a discussion that integrates relevant literature, emphasizing the environmental and health implications of trace element bioaccumulation. These revisions have substantially strengthened the scientific value of the study.

In its current form, the manuscript provides a valuable contribution to the field of environmental and health risk assessment related to trace elements in fish. I recommend its acceptance for publication in PLOS ONE.

7. PLOS authors have the option to publish the peer review history of their article (what does this mean?). If published, this will include your full peer review and any attached files.

Reviewer #2: No

Reviewer #4: **Yes: **Mahmoud Mahrous M. Abbas

---

## [Editor Report · Acceptance letter]

6 Jan 2025

PONE-D-24-39090R1 

PLOS ONE

Dear Dr. Ullah, 

I'm pleased to inform you that your manuscript has been deemed suitable for publication in PLOS ONE. Congratulations! Your manuscript is now being handed over to our production team.

Kind regards, 

on behalf of

Dr. Ulaganathan Arisekar 

Academic Editor

PLOS ONE